**EMBO *reports***

# Complex interplay between RAS GTPases and RASSF effectors regulates subcellular localization of YAP

Swati Singh [ID][1], Gabriela Bernal Astrain[1], Ana Maria Hincapie[2], Marilyn Goudreault[1] &
Matthew J Smith [ID][1,3]✉

## Abstract

RAS GTPases bind effectors to convert upstream cues to changes in cellular function. Effectors of classical H/K/NRAS are defined by RBD/RA domains which recognize the GTP-bound conformation of these GTPases, yet the specificity of RBD/RAs for over 160 RAS superfamily proteins remains poorly explored. We have systematically mapped interactions between BRAF and four RASSF effectors, the largest family of RA-containing proteins, with all RAS, RHO and ARF small GTPases. 39 validated complexes reveal plasticity in RASSF binding, while BRAF demonstrates tight specificity for classical H/K/NRAS. Complex between RASSF5 and diverse RAS GTPases at the plasma membrane can activate Hippo signalling and sequester YAP in the cytosol. RASSF8 undergoes liquid-liquid phase separation and resides in YAP-associated membraneless condensates, which also engage several RAS and RHO GTPases. The poorly studied RASSF3 has been identified as a first potential effector of mitochondrial MIRO proteins, and its co-expression with these GTPases impacts mitochondria and peroxisome distribution. These data reveal the complex nature of GTPase-effector interactions and show their systematic elucidation can reveal completely novel and biologically relevant cellular processes.

**Keywords** RAS GTPase; RASSF; Hippo Pathway; YAP; MIRO
**Subject Category** Signal Transduction

## Introduction

Small GTPases are a class of critical hub proteins that control a diverse array of cellular processes. There are ~160 proteins in the RAS superfamily that cluster into five groups: RAS, RHO, RAN, RAB, and ARF (Colicelli, 2004). The RAS subfamily consists of 35 small GTPases, some of which promote cell growth and proliferation (RAS, RAL, RIT, RHEB, and RAP) while others induce cell cycle arrest and/or apoptosis more consistent with a tumor

suppressor function (RASD, NKIRAS, RASL, DIRAS, and RGK GTPases) (Bernal Astrain et al, 2022). Three RAS proteins (HRAS, KRAS, NRAS; hereon H/K/NRAS or classical RAS) have been studied extensively, attributed to their role in oncogenesis and frequent activation in human cancers (Hobbs et al, 2016). Research into most alternative RAS subfamily GTPases remains limited. RAS GTPases bind downstream effector proteins when in a GTP-loaded, activated conformation (Kiel et al, 2021). Engagement is driven by interaction with effector RAS Binding Domains (RBDs) or RAS Association (RA) domains, small protein modules sharing a ubiquitin-like fold. There are over 50 proteins with RBD/RA domains in the human proteome and most are still considered direct effectors of H/K/NRAS (Smith, 2023; Singh and Smith, 2020). The few data available, however, suggest these domains do not universally associate with H/K/NRAS: phosphoinositide-3-kinases-β (PI3Kβ) does not complex with RAS subfamily GTPases but rather RAC1 and CDC42 (Fritsch et al, 2013), seven distinct RAS proteins bind and activate PI3Kα (Yang et al, 2012), and an RBD from the RAC regulator ELMO (Engulfment and cell motility) binds to RHO and ARF GTPases (Patel et al, 2011).

The ten-member RAS association domain family (RASSF) constitutes nearly 20% of the known RBD/RA-containing effector proteins. They are simple scaffolds that can be grouped into two categories: classical C-RASSFs (RASSF1-6) with an RA towards the C-terminus followed by a helical SARAH (Salvador/Rassf/Hippo) motif, and N-RASSFs (RASSF7-10) that have an RA at the N-terminus followed by a coiled-coil region (Pfeifer et al, 2010). All 10 RASSFs were long considered H/K/NRAS effectors, but recent work revealed only RASSF5 directly binds the classical RAS proteins, leaving the remaining nine RASSF proteins with no known GTPase-binding partner (Dhanaraman et al, 2020). Analyses of RASSF1 specificity towards several GTPases of the RAS subfamily found this protein complexes with the non-canonical GTPases GEM, REM1, REM2, RASL12, DIRAS3, and ERAS (Dhanaraman et al, 2020). The biological significance of these interactions is still being investigated, but all RASSF proteins appear to function as tumor suppressors in a wide variety of human cancers, whereby the promoters of *RASSF* genes are frequently hypermethylated to inhibit their expression (Richter et al, 2009). *RASSF1A* is downregulated in >80% of lung cancers and mice lacking *RASSF1A* are prone to tumourigenesis (Dammann et al,

[1]Institute for Research in Immunology and Cancer, Université de Montréal, Montréal, QC H3T 1J4, Canada. [2]Rosalind and Morris Goodman Cancer Institute, McGill University, Montréal, QC H3A 1A3, Canada. [3]Department of Pathology and Cell Biology, Faculty of Medicine, Université de Montréal, Montréal, QC H3T 1J4, Canada.
✉E-mail: matthew.james.smith@umontreal.ca

2000; Tommasi et al, 2005). The exact mechanism by which RASSF effectors constrain cell growth and proliferation have not been elucidated, but the proteins have been linked to Hippo signaling, apoptosis, cell cycle regulation, microtubule and genomic stability (Dhanaraman et al, 2020; Baksh et al, 2005; Shivakumar et al, 2002; Singh and Smith, 2020). Loss of *RASSF5*, the only direct effector of H/K/NRAS, cooperates with activated KRAS oncoproteins to drive cellular transformation and tumor growth (Park et al, 2010). In cultured cells, RASSF5 was shown to activate the Hippo pathway in a KRAS-dependent manner (Dhanaraman et al, 2020).

The Hippo pathway is an evolutionary conserved regulator of growth and proliferation that responds to cues from cell adhesion, polarity and mechanical forces to control tissue size, homeostasis and regeneration (Zheng and Pan, 2019). The core of the pathway comprises a kinase cascade wherein the serine/threonine kinases MST1/2 (orthologs of *Drosophila* hippo/Hpo), along with their partner SAV phosphorylate and activate LATS1/2 and their co-factor MOB1A/B (Yu and Guan, 2013; Chan et al, 2005). Active LATS1/2 phosphorylates the transcriptional co-activators YAP/TAZ, which are sequestered in the cytoplasm *via* 14-3-3 proteins and subject to degradation (Hao et al, 2008; Zhang et al, 2010). When Hippo kinases are inactive, YAP remains nuclear and associates with transcription factors such as TEA domain proteins 1–4 (TEAD) to drive the expression of growth-promoting genes (Wu et al, 2008). Upstream of this pathway is an array of positive and negative regulators, including the striatin-interacting phosphatase and kinase (STRIPAK) complex, a negative regulator that inactivates MST1/2 by dephosphorylation, as well as upstream activators such as angiomotins (AMOT, AMOTL1, and AMOTL2), KIBRAs (KIBRA, WWC2, and WWC3), Neurofibromatosis-2 (NF2) and protein tyrosine phosphatase nonreceptor type 14 (PTPN14) (Yu and Guan, 2013). Deregulated expression of Hippo pathway genes is observed in numerous human cancers (Harvey et al, 2013; Piccolo et al, 2023). *YAP* is a recognized protooncogene and the Hippo activators NF2, MST1/2, and LATS1/2 are tumor suppressors. The helical SARAH motifs of (C-)RASSF1-6 hetero-dimerize with SARAH domains of the MST1/2 kinases, but their role in regulating Hippo signaling has been poorly explored (Hwang et al, 2007). (N-)RASSF7-10 lacks a SARAH motif and a role in Hippo signaling is not established, yet interactome data suggests these effectors are also associated with Hippo proteins (Hauri et al, 2013).

Though RASSFs have gathered significant interest accredited to their downregulation in human cancers, knowledge of their biological function remains limited. Here, we systematically map (C-)RASSF3, RASSF4, RASSF5, and (N-)RASSF8 interactions with all RAS, RHO, and ARF GTPases. The data reveal that not all proteins with RBD/RA domains are effectors of classical RAS GTPases. We can detect Hippo pathway activation by multiple RAS subfamily GTPases in a manner dependent on RASSF5. Furthermore, the approach uncovered the first identified small GTPase partners for RASSF3, RASSF4, and RASSF8, including many small G-proteins with no previously known effectors. This includes the mitochondrial GTPases MIRO1/2 (RHOT1/2) which bind directly to RASSF3 and recruit this protein to mitochondria. We provide evidence that RASSF8 resides in liquid-like phase-separated condensates, similar to other Hippo components, and these are dynamically regulated by an array of novel GTPase partners. This study exposes the increasing complexity of effector-GTPase interactions and the biological significance of RASSF proteins, which are amongst the most downregulated tumor suppressors in human cancers.

# Results

## Mapping RAS, RHO, and ARF GTPase interactions with effector RBD domains

In a previous study we reported that only RASSF5 binds to H/K/NRAS, leaving the remaining 9 RASSF effectors with no known GTPase partner (Dhanaraman et al, 2020). A computational approach identified the RGK GTPases (REM1, REM2, and GEM) as interacting partners for RASSF1, but this is not applicable as a general methodology to identify GTPase-effector pairs. To better understand the specificity of RBD/RA domains and identify binding partners for diverse RASSF proteins, we attempted to map RASSF interactions with all RAS, RHO, and ARF subfamily GTPases. The use of discovery-based approaches (e.g., proteomics) to comprehensively identify small GTPase interacting partners for putative effectors is constrained by the absolute requirement to have the proteins in a GTP-loaded state, and the differential expression profiles of GTPases across cell and tissue types. Thus, to map RASSF interactions with small GTPases we took a systematic approach based on precipitation of mutationally activated GTPases by purified RBD/RA domains. Pull-down assays using recombinant, GST-tagged domains were performed using Bio-Spin columns and a vacuum manifold to increase throughput (Fig. 1A). Glutathione beads and RBD/RA domains were mixed with HEK 293T whole cell lysates expressing VENUS-tagged, constitutively active variants of all RAS (35), RHO (22), and ARF (27; excluding ARL9, ARL16, and ARL17) subfamily GTPases. RASSF homologs cluster into three groups based on sequence homology: RASSF1/3/5, RASSF2/4/6, and RASSF7-10 (Fig. 1B). We probed the GTPase-binding landscape for at least one RASSF from each cluster, specifically (C-)RASSF3, RASSF4, RASSF5, and (N-)RASSF8. We included the RBD of BRAF to compare its interaction profile with that of RASSF domains, as RAF kinases are highly studied effectors that bind KRAS with nanomolar affinity (Herrmann et al, 1995, 1996). GST alone served as a control to identify non-specific interactions. Each experiment included positive and negative controls, specifically RASSF5 binding to activated KRAS-G12V or CDC42-G12V, respectively (Fig. 1C; blots from full screens are in appendix Figs. S1–6). Binding intensities observed for each RBD/RA-GTPase interaction on Western blots were used to generate a heatmap representing effector interactions with all GTPases, with red representing a strong interaction, yellow moderate binding, and black no interaction (heatmap for RAS subfamily in Fig. 1D, for RHO and ARF in Fig. EV1A,B). The BRAF RBD showed robust and specific interaction with H/N/KRAS, demonstrating selectivity for the classical RAS GTPases. BRAF did not strongly associate with any of the 80 other RAS, RHO, or ARF GTPases. This attests that BRAF is a highly specific H/K/NRAS effector, consistent with its nanomolar affinity for the GTP-loaded conformation of these GTPases.

In contrast to RAF, the RASSF5 effector has a moderate affinity for KRAS ($K_d$ 1.7 μM (Dhanaraman et al, 2020)), suggesting it may not have the same restricted specificity. Indeed, the RASSF5 RA bound multiple RAS subfamily GTPases, including H/K/NRAS and seven related proteins (RRAS, RRAS2, MRAS, RAP2B, RAP2C, RIT1, and RIT2; Fig. 1D). There is an apparent functional diversity

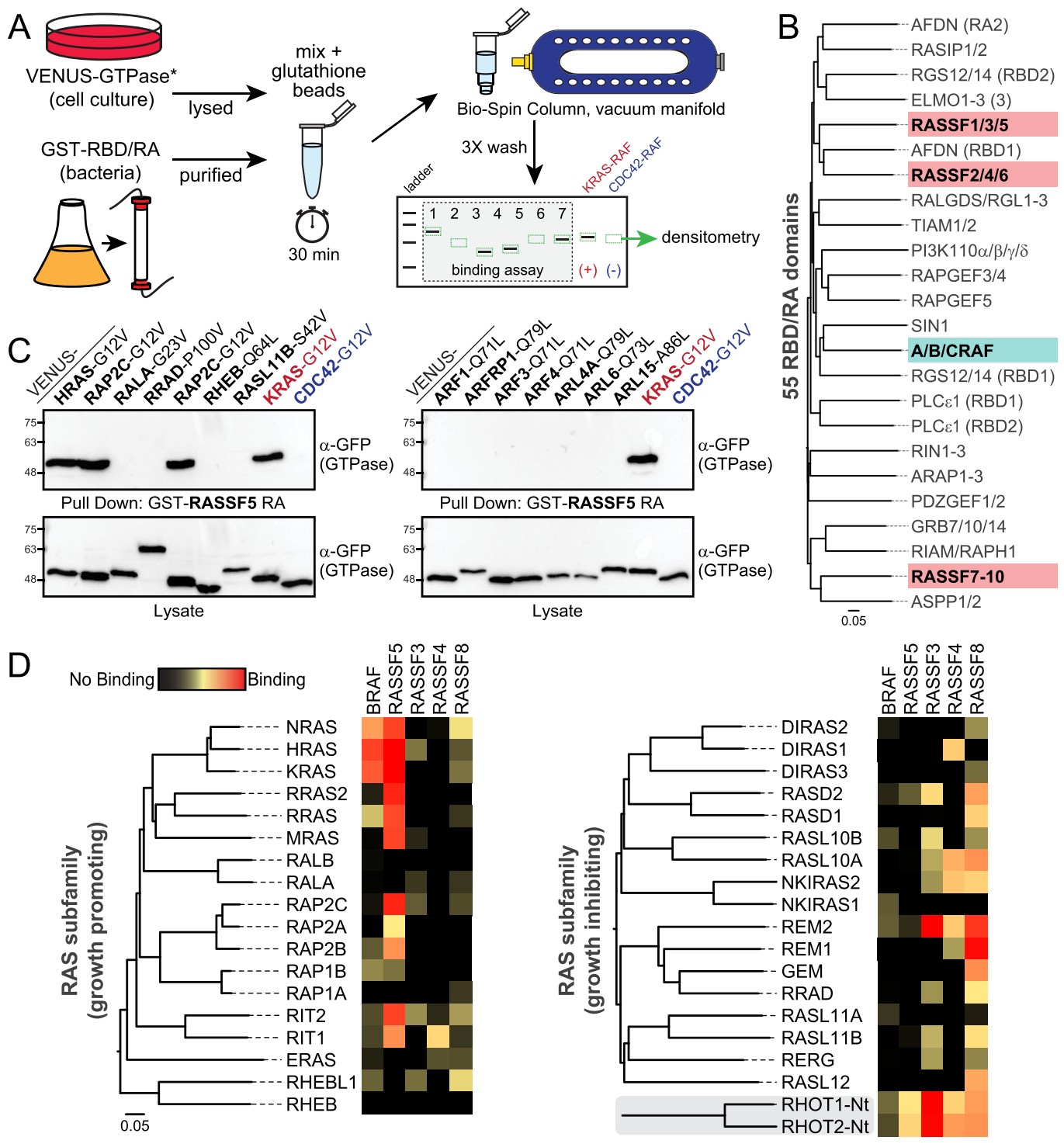

**Figure 1. Systematic screen to map RAS, RHO, and ARF GTPase interactions with BRAF RBD and RASSF RA domains.**

(A) Schematic showing the experimental approach to perform GST-RBD/RA precipitation of VENUS-tagged, mutationally activated (*) GTPase assays in a high-throughput manner. (B) Dendogram of 55 RBD/RA domains from putative RAS effectors in the human proteome. The ten RASSF homologs are present in three unique clusters (RASSF1/3/5, RASSF2/4/6, and RASSF7/8/9/10) highlighted in red. RAF kinase RBDs are highlighted in blue. (C) Representative image of Western blots showing precipitation of VENUS-GTPases by GST-tagged RASSF5 RA following precipitation. KRAS-G12V with RASSF5 was included as a positive control on each blot, and CDC42-G12V served as a negative control. (D) Heatmap of BRAF and RASSF5/3/4/8 interactions with RAS subfamily GTPases. The subfamily is divided into putative growth-promoting GTPases (left) and those with tumor suppressor properties (right). The MIRO1/2-nGTPase domains cluster close to the RAS subfamily. Strong interactions are in red, moderate in yellow, and no interaction in black. Heatmap is generated from two biological replicates of pull-down assays. Source data are available online for this figure.

in the 35-member RAS subfamily, with 18 GTPases promoting growth and proliferation while the remaining 17 have growth suppressive properties (Bernal Astrain et al, 2022). Interestingly, the RA of the tumor suppressor RASSF5 interacted only with growth-promoting RAS GTPases. There were no strong hits in the RHO subfamily and only weak interactions with ARL4C and ARL11 of the ARF subfamily (Fig. EV1A,B). The RBDs of RASSF3, RASSF4, and RASSF8 do not bind classical RAS proteins and had no known GTPase partners. In our screen, no interaction was observed between any of these effectors and growth-promoting GTPases from the RAS subfamily. Instead, several growth inhibitory RAS GTPases were identified as the first interacting partners for RA domains of RASSF3, RASSF4, and RASSF8 (Fig. 1D). RASSF3 hits included RASD2, RASL10B and REM2. The RASSF4 RA interacted with DIRAS1, RASL10A, NKIRAS2 and REM2, while RASSF8 precipitated RASD1, RASD2, RASL10A, NKIRAS2, REM1, REM2, GEM, RRAD, RASL11B, and RASL12. The other prominent and common interacting GTPase partners for the four RASSF homologs were nGTPase domains from MIRO1 (also RHOT1) and MIRO2 (RHOT2). Initially described as RHO-like, MIRO GTPases cluster by sequence similarity between the RAS and RHO subfamilies but lack the RHO helical insert. They are tethered to mitochondria and comprise two GTPase domains, one N-terminal (nGTPase) and another C-terminal (cGTPase), separated by a pair of calcium-binding EF-hands (Klosowiak et al, 2013). Among the four RASSF domains, the RA of RASSF3 showed the strongest binding to MIRO1/2. Finally, from the RHO subfamily, we observed RASSF3, RASSF4, and RASSF8 binding to RHOBTB1, RHOBTB2, and RHOH (Fig. EV1A). RASSF3 and RASSF4 also complexed with RHOF. Few of the ARF subfamily GTPases were identified as potential interacting partners, though ARL4C interacted with all four of the RASSF RA domains (Fig. EV1B) but not the GST alone control. ARL11 and ARL5C also precipitated with RASSF5, RASSF3 and RASSF8. RASSF8 bound ARL5B, ARL3, ARL13B, ARL10, and ARL15. Thus, using a systematic approach for screening GTPase-effector interactions, we successfully generated a large and comprehensive map of BRAF and RASSF interactions with 83 GTPases of the RAS, RHO, and ARF subfamilies. The data reveal distinct binding of individual RA domains and high BRAF selectivity for the classical RAS GTPases. The RASSF effectors have increased plasticity and an apparent capacity to signal from an array of small GTPases, predominantly from the RAS subfamily.

## Multiple RAS subfamily GTPases activate Hippo signaling through RASSF5

RASSF5 is a direct binding partner of the mammalian Hippo kinases and stimulates their activity downstream of KRAS. Our screen identified 10 RAS subfamily GTPases that could potentially trigger Hippo activation through RASSF5, including the three classical RAS proteins. To validate these interactions we first used co-precipitation of activated VENUS-tagged GTPases expressed in HEK 293T cells. These could be precipitated by recombinant GST-RASSF5 RA, as in the initial screen, and not by a GST control (Fig. EV1C). When co-transfected with full-length FLAG-tagged RASSF5, all 10 candidate GTPases were co-immunoprecipitated (co-IP; Fig. EV1D). An activated CDC42 variant served as a negative control and did not complex with RASSF5. The use of

constitutively active (GTP) or dominant-negative (GDP) variants revealed that RASSF5 interacts with these GTPases in a nucleotide-dependent manner. Recombinant GST-RASSF5 RA strongly bound active mutants of KRAS (G12V or Q61L), RRAS (G30V or Q87L), and RIT1 (G30V or Q79L) as well as RIT1 wild-type (WT). However, RASSF5 RA showed no binding to dominant-negative mutant of KRAS (S17N) and RIT1 (S35N), and diminished binding to KRAS WT, RRAS WT and RRAS S27N (Fig. 2A). No GTPases were precipitated by GST-alone. To further corroborate these interactions, we assayed whether activated GTPases could alter the subcellular localization of RASSF5. In human epithelial HeLa cells, mCherry-RASSF5 is prominent in the nucleus and to a lesser extent the cytosol (Fig. 2B). Upon co-expression with the classical RAS GTPases, RASSF5 showed complete redistribution to the plasma membrane (PM) (Fig. 2B). Other hits included the GTPases RRAS and RRAS2, MRAS, and RAP2B/C, which each comprise a CaaX motif (where C=Cys, a=aliphatic residue, and X=any amino acid) at their C-terminus suggesting PM tethering. RAP2 GTPases have also been found in endosomes (Ohba et al, 2000; Uechi et al, 2009). Co-expression of RASSF5 with any of these GTPases redistributed RASSF5 to the PM (Fig. 2C). RIT1 and RIT2 lack a canonical CaaX motif, though we observed RIT1 at the PM as described previously (Cuevas-Navarro et al, 2023). When co-expressed with RASSF5, RIT1 could recruit this effector to the membrane (Fig. 2D). RIT2 was prominently localized in the nucleus and is co-expression with RASSF5 retained RASSF5 in the nucleus with no evidence of cytosolic RASSF5. Thus, all ten candidate RAS subfamily GTPases from the screen could both complex with RASSF5 and recruit this effector to specific subcellular locales.

Though initially deemed "death effectors" thought to promote apoptosis, previous work has shown that RASSFs lack an intrinsic ability to stimulate apoptosis even in the presence of KRAS (Dhanaraman et al, 2020). Instead, RASSF5 co-expression with KRAS blocks nuclear localization of the transcriptional co-activator YAP, suggesting RASSF5 couples RAS activation to the Hippo pathway. We explored whether RASSF5 and its newly identified GTPase partners may universally control Hippo activation. We first sought to determine whether endogenous RASSF protein levels would be sufficient to study signaling. Antibodies that recognize exogenously expressed RASSF5 were unable to detect significant levels of endogenous protein in U2OS, HeLa, HEK, or HCT 116 cells (Fig. 3A). Similarly, no endogenous RASSF3 or RASSF8 were detected in U2OS cells, a human bone osteosarcoma epithelial line that is highly efficacious to study Hippo signaling. We could, therefore, use U2OS cells to assay signaling through individual RASSF effectors using transient expression. To investigate whether RASSF5 interaction with distinct RAS subfamily GTPases can influence Hippo signaling, we stained for endogenous YAP in sparsely confluent U2OS cells. In the absence of activated GTPases or RASSF5, Hippo is inactive and YAP resides in the nucleus. Transient expression of RASSF5 did not alter the nuclear localization of YAP (Fig. 3B), indicating that RASSF5 alone does not stimulate Hippo activity. Similarly, individual expression of the 10 constitutively active RAS subfamily GTPases did not affect nuclear YAP (Figs. 3B and EV2A). Co-expression of RASSF5 with any of the PM-localized RAS GTPases (H/N/KRAS, RRAS/2, MRAS, RAP2B/2C, and RIT1), however, clearly resulted in YAP redistribution to the cytosol (Fig. 3C and EV2B). This suggests the interaction between RASSF5 and its RAS subfamily partners results

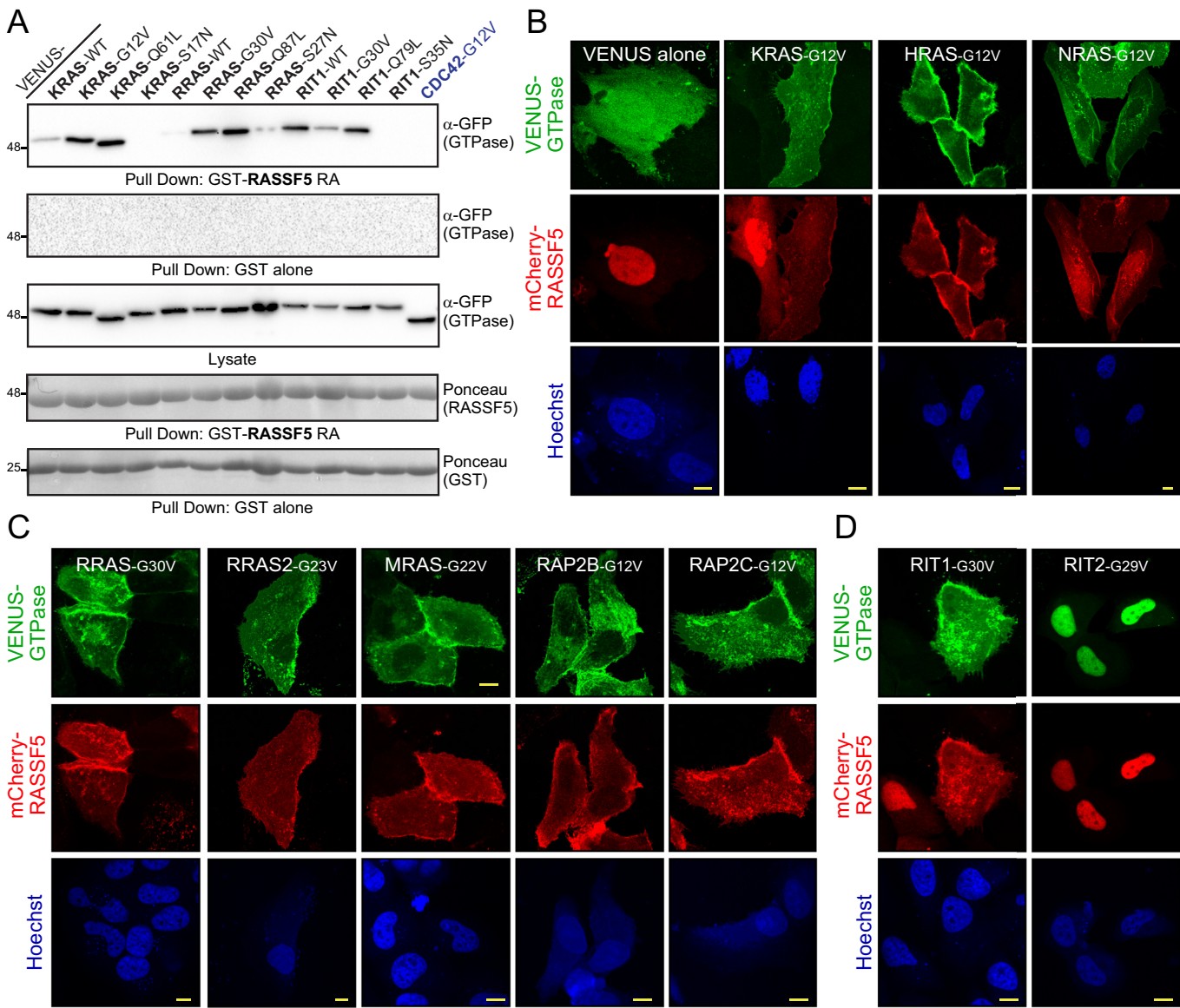

**Figure 2. RASSF5 interacts with numerous distinct RAS subfamily GTPases.**

(A) Pull-down assays with recombinantly purified GST-RASSF5 RA domain and variants of VENUS-tagged KRAS (wild-type (WT), constitutively active (G12V and Q61L) and dominant negative (S17N)) together with corresponding mutants of RRAS and RIT1 expressed in HEK 293T cells. CDC42, as well as pull-down with GST alone, served as negative controls. (B) Representative confocal microscopy images showing subcellular localization of mCherry-RASSF5 when expressed with VENUS alone (left) or VENUS-tagged H/K/NRAS GTPases (mutationally activated) in HeLa cells. The nucleus was detected with Hoechst and is shown in blue. Scale bars represent 10 μm. (C) Confocal images of HeLa cells co-expressing mCherry-RASSF5 and candidate VENUS-tagged, mutationally activated RAS-related GTPases. Scale bars represent 10 μm. (D) Representative images of VENUS-tagged RIT1 and RIT2, which both lack a canonical CaaX motif, co-expressed with mCherry-RASSF5 in HeLa cells. Scale bars represent 10 μm. All experiments were performed in three independent biological replicates. Source data are available online for this figure.

in the activation of Hippo signaling and the removal of YAP from the nucleus. Interestingly, this was not observed when RASSF5 was expressed with the CaaX-less GTPase RIT2 (Fig. 3D). Moreover, truncating the SARAH domain of RASSF5, which prevents binding to the MST kinases, also blocked stimulation of Hippo activity downstream of activated KRAS (Figs. 3E and EV2C). These data demonstrate that multiple GTPases from the RAS subfamily can activate the Hippo pathway in a manner dependent on RASSF5 and demonstrate the importance of RASSF5 localization with GTPases at the PM in activating Hippo.

## RASSF8 forms dynamic LLPS condensates in cells

H/K/NRAS are highly studied, attributed to their role in cancer, while research on many alternative RAS GTPases remains limited. Data describing a biological function for RASSF8 are also scarce, with RASSF1 and RASSF5 receiving the most consideration to date. To determine a biological significance for the novel RASSF8-GTPase interactions identified in our screen we first examined the subcellular localization of RASSF8. *Drosophila* dRASSF8 was reported at adherens junctions in vivo, but the localization of

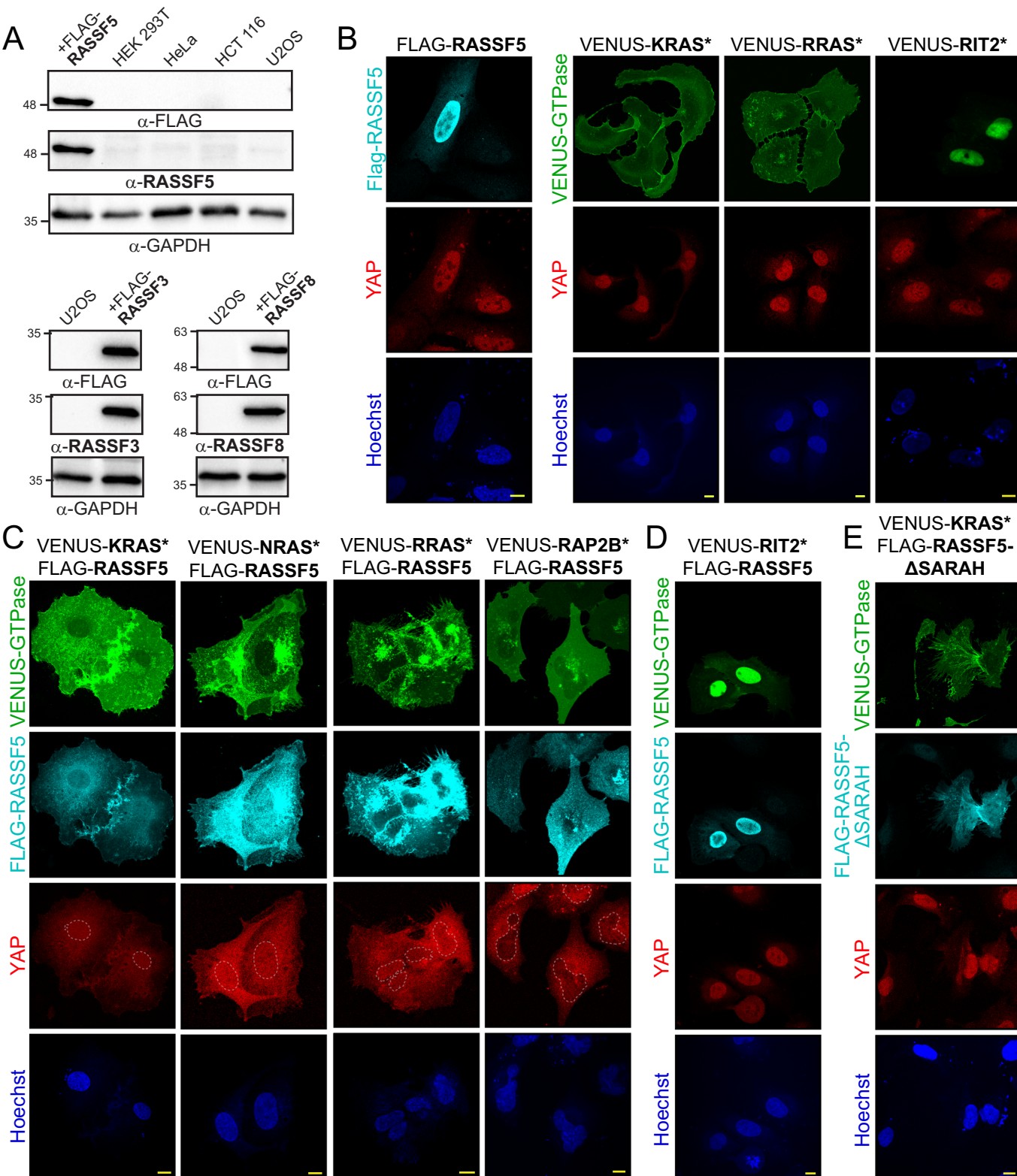

RASSF8 in cultured cells was unknown (Langton et al, 2009; Zaessinger et al, 2015). Consistent with its absence in Western blots (Fig. 3A), immunostaining for RASSF8 was not successful. Transiently expressed RASSF8 was consistently present in punctate-like structures in the cytosol (Fig. EV3A). This was observed when RASSF8 was tagged with mCherry, VENUS or FLAG. mCherry-RASSF8 showed a high number of foci and was co-localized with FLAG-RASSF8 when the recombinant proteins were co-expressed. We investigated whether RASSF8 might co-localize with subcellular structures that have a punctate

**Figure 3.   Multiple RAS subfamily GTPases activate Hippo signaling through RASSF5.**

(A) Expression levels of RASSF3, RASSF5, and RASSF8 in several cultured cell lines as detected by antibodies against the endogenous proteins. Expression of exogenous, FLAG-tagged proteins in U2OS cells confirmed the antibodies specifically recognize the appropriate target. GAPDH served as a loading control. (B) Immunofluorescence images of endogenous YAP (red) in sparsely confluent U2OS cells transiently expressing either FLAG-RASSF5 (cyan, left) or the mutationally activated (*) VENUS-GTPases KRAS-G12V, RRAS-G30V, or RIT2-G29V. Nuclei are visualized with Hoechst staining (blue). Scale bars represent 10 μm. (C) Immunostaining of endogenous YAP (red) in sparsely confluent U2OS cells co-transfected with FLAG-RASSF5 (cyan) and mutationally activated (*) RAS GTPases (KRAS-G12V, NRAS-G12V, RRAS-G30V, RAP2B-G12V). Hoechst staining (blue) was used to visualize nuclei, and scale bars represent 10 μm. (D) Endogenous YAP (red) in U2OS cells co-expressing FLAG-RASSF5 (cyan) and mutationally activated (*) VENUS-RIT2-G29V. Hoechst (blue) was used to detect nuclei. Scale bars represent 10 μm. (E) Immunostaining for endogenous YAP (red) in U2OS cells expressing FLAG-RASSF5-ΔSARAH (cyan) and constitutively activated (*) VENUS-KRAS-G12V. Scale bars represent 10 μm. Data were representative of n = 3 independent experiments. Source data are available online for this figure.

distribution. mCherry-RASSF8 foci did not co-localize with endosomes (marked with anti-EEA1) or autophagosomes (LC3) (Fig. 4A). Further, RASSF8 did not co-distribute with stress granules (G3BP1), mitochondria (MitoTracker), golgi (GM130) or trans-golgi (GOLGIN97) (Fig. EV3B). We then considered whether the observed puncta might be membraneless organelles established through liquid-liquid phase separation (LLPS), which would explain the circular pattern observed when immunostaining with anti-FLAG (Fig. EV3A). Current requirements to demonstrate LLPS-like properties in cells include demonstrating that puncta are dynamic using fluorescence recovery after photobleaching (FRAP), and observations of puncta fusion. In time-lapse live imaging of U2OS cells expressing mCherry-RASSF8, we observed fast and near-complete fluorescence recovery within 2 min of photobleaching (Figs. 4B and EV3C). We also observed that puncta were dynamic and could fuse with nearby foci. These data strongly suggest that RASSF8 forms liquid-like phase-separated condensates, or membraneless organelles.

Recent studies have reported that many proteins of the Hippo pathway, including the upstream regulators AMOT and KIBRA, the Hippo kinases MST1/2 and LATS1/2, and the downstream transcriptional co-activators YAP/TAZ all form phase-separated condensates (Wang et al, 2022; Bonello et al, 2023). In the case of AMOT and KIBRA, LLPS is mediated by long coiled-coil (cc) domains (Wang et al, 2022). Interestingly, mining published proteomics and genomics data suggests RASSF8 is also involved in Hippo signaling (Fig. EV3D,E) despite lacking the MST-binding SARAH motif found in the C-RASSFs. (N)-RASSF8 does comprise a predicted coiled-coil domain in its C-terminal half (Fig. 4C), and we first examined whether the formation of RASSF8 foci is driven by this domain. Indeed, truncating the coiled-coil (RASSF8-Δcc) resulted in the complete elimination of the punctate-like structures compared to full-length RASSF8 (Fig. 4D,E), establishing that RASSF8 condensates are mediated by its coiled-coil region and not the GTPase-binding RA domain. To further investigate a connection with Hippo signaling we explored whether sorbitol treatment, an inducer of osmotic stress and activator of Hippo, could promote RASSF8 puncta formation. We found that a 2 min exposure to 0.4 M sorbitol rapidly increased the number of mCherry-RASSF8 condensates (Fig. 4F) with complete penetrance. Upon removing sorbitol, the number of condensates returned to pre-treatment levels. No condensates were detected upon sorbitol treatment of cells expressing mCherry-RASSF8-Δcc. We could also corroborate the rapid induction of YAP condensates by sorbitol as observed previously (Fig. 4F). Co-expression of EGFP-YAP and mCherry-RASSF8 revealed that YAP and RASSF8 condensates co-localize in the cytosol (Fig. 4G). We did not observe the co-localization of

RASSF8 with YAP condensates in the nucleus. mCherry-RASSF8-Δcc did not co-localize with YAP, suggesting that RASSF8 LLPS and YAP association are driven by its coiled-coil region. Taken together, these data support a role for RASSF8 in Hippo signaling, mediated by a coiled-coil domain disposed to LLPS and association with YAP.

## RASSF8 complex with tumor suppressor-class RAS GTPases

The distribution of RASSF8 to membraneless organelles through its coiled-coil domain should leave the RA domain accessible for target engagement. Our screen identified multiple candidate GTPase partners for RASSF8, primarily tumor suppressor-class RAS subfamily GTPases for which there are little functional data and several RHO subfamily members. We confirmed that full-length RASSF8 can complex with these RAS and RHO GTPases by co-IP (FLAG-RASSF8 with RASD1/2, RASL10A/B, RASL11B, RASL12, NKIRAS2, the RGK GTPases (GEM, REM1, REM2, and RRAD) and the RHO GTPases RHOBTB1/2, RAC2 and RHOH) (Fig. EV4A). As in the pull-down screen with RA domain, KRAS did not co-precipitate with full-length RASSF8. There are no canonical GEFs or known activating conditions for the majority of these small GTPases, leaving the use of the mutationally activated variants as the only mechanism to study their function in a presumed GTP-loaded state. An exception is the RHOBTB proteins, which are pseudoGTPases that are constitutively GTP-bound (Stiegler and Boggon, 2020). Transiently expressing six potential RASSF8 partners showed that VENUS-tagged GTPases were generally distributed in the cytosol, with minimal distribution to small puncta (Fig. 5A). Upon co-expression with RASSF8 we observed that, instead of multiple punctate structures around the cytosol, RASSF8 and the RAS subfamily GTPases RASD1/2, RASL11B, and RASL12 co-localize in large perinuclear structures (Fig. 5B). For RASD2 and RASL12 this was dependent on mutational activation, as wild-type GTPases did not co-localize with RASSF8 (Fig. EV4B). Wild-type RASD1 and RASL11B mostly co-localized to the large structures, though it is possible these are GTP-loaded in cells as their nucleotide cycling has never been elucidated. Co-expression of wild-type RHOBTB1/2 with RASSF8 also revealed co-localization to intense, punctate-like structures (Fig. 5B). Thus, 15 candidate small GTPases of the RAS and RHO subfamilies can complex with RASSF8, and many of these co-localize with this effector in large structures around the perinuclear region or throughout the cytosol.

Finally, we asked whether YAP was contained in the perinuclear structures containing RASSF8 and its partner GTPases, or whether

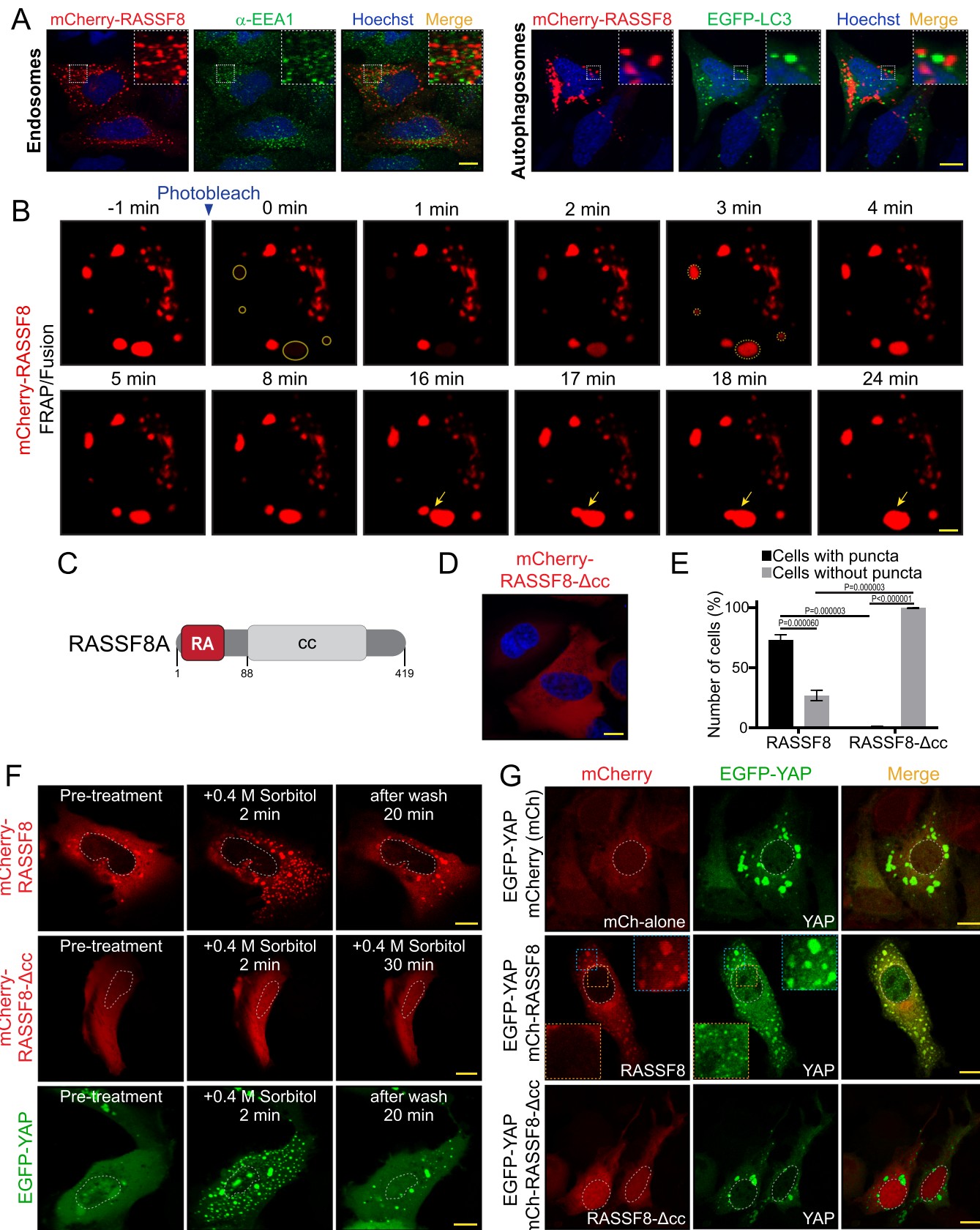

Figure 4. RASSF8 and YAP form biomolecular condensates in cells via LLPS.

(A) Fluorescence images of U2OS cells transiently expressing mCherry-RASSF8. These were stained with a marker for endosomes (anti-EEA1, left) or were co-transfected with a marker for autophagosomes (EGFP-LC3, right). Nuclei are in blue. Scale bars represent 10 µm. (B) Time-lapse live-cell imaging of U2OS cells expressing mCherry-RASSF8 demonstrates rapid FRAP and fusion of distinct foci. Encircled puncta (yellow) were photobleached and exhibited fluorescence recovery within 2 min. Arrows (yellow) show the fusion of two distinct condensates. Time is indicated above panels and scale bar represents 10 µm. (C) Domain architecture of RASSF8 shows the positioning of the RA domain and coiled-coil region. (D) Representative image of mCherry-RASSF8-Δcc in U2OS cells. The nucleus is visualized with Hoechst (blue). The scale bar represents 10 µm. (E) Quantification of individual U2OS cells with or without puncta following transient expression of mCherry-RASSF8 or mCherry-RASSF8-Δcc. Data represent mean ± SD, $n = 3$ biological replicates with >150 cells for each, multiple unpaired $t$-test. $P$ values are marked on the plot, all were determined significant at $P < 0.0001$. (F) Sorbitol treatment of U2OS cells expressing mCherry-RASSF8, -RASSF8-Δcc, or EGFP-YAP. Live-cell imaging shows pre- and post-treated cells (2 min) as well as post-washing of sorbitol. Nuclei are marked with dashed lines. Scale bars represent 10 µm. Data were representative of $n = 3$ independent experiments.
(G) Confocal microscopy images of fixed U2OS cells co-expressing EGFP-YAP and either mCherry alone, mCherry-RASSF8, or mCherry-RASSF8-Δcc. Nuclei are marked with dashed lines. Scale bars represent 10 µm. Data were representative of $n = 3$ independent experiments. Source data are available online for this figure.

these complexes were mutually exclusive with RASSF8-YAP association. Upon co-expression of ECFP-tagged GTPases (RASD1, RASL11B, and RHOBTB1) with mCherry-RASSF8 and EGFP-YAP, we observed complete coalescence of these proteins in perinuclear structures (Fig. 5C). Control experiments ensured no bleed through from the individual channels (Fig. EV4C). We also observed that co-expression of EGFP-YAP with the small GTPases alone, in the absence of RASSF8, drove coalescence of YAP condensates with GTPases in perinuclear structures (Fig. 5D). It is possible this is mediated through alternative N-RASSF proteins, or by the related ASPP1/2 effectors which bind N-RASSF proteins (Dhanaraman et al, 2020). In time-lapse live imaging of U2OS cells expressing mCherry-RASSF8 and VENUS-tagged, mutationally activated RASD1 or RHOBTB1-WT, RASSF8 condensates showed a much-reduced fluorescence recovery after photobleaching characteristic of a gel-like state (Fig. EV4D,E). This suggests that the coalescence of RASSF8 and GTPase proteins in perinuclear structures changes the dynamics of RASSF8 condensates and drives a liquid-to-solid phase transition. These data indicate that multiple GTPases of the growth inhibitory RAS subfamily, along with the known Hippo regulator RHOBTB1, are present in RASSF8-YAP condensates and interaction with these GTPases can mediate liquid-to-solid phase separation of RASSF8.

## RASSF3 is a direct effector of the mitochondrial GTPases MIRO1 and MIRO2

Outside of their recognition as tumor suppressors there are few functional data on RASSF3 or RASSF4. Both showed interaction with multiple candidate GTPases of the RAS, RHO, and ARF subfamilies in our screen, predominantly with tumor suppressor-class RAS subfamily GTPases. We could confirm that full-length RASSF3 and RASSF4 interact with these GTPases using co-IP of FLAG-tagged RASSFs and mutationally activated GTPases. RASSF3 complexed with the RAS proteins REM1/2, RASD1/2, RASL10A, and RASL11B, as well as the nGTPase domains of the mitochondrial GTPases MIRO1/2 (Fig. 6A). RASSF4 also complexed with MIRO1/2-nGTPase, and the RAS GTPases REM1/2, RASL10A, and NKIRAS2. An analysis of known RASSF3 interactors revealed a clear relationship with Hippo signaling, particularly the STRIPAK complex and the Hippo kinases MST1/2 (Fig. EV5A). Additional partners include the microtubule-binding proteins MAP1B and MAP1S, the mitochondrial proteins PGAM5 and FGFR1OP2, and endoplasmic reticulum proteins VAPA and VAPB. This indicated RASSF3 may function in conjunction with

organelle-tethered proteins and microtubules, a clear connection to the MIRO GTPases which are present on the outer mitochondrial membrane and interact with microtubule-binding proteins. Unlike typical RAS GTPases, MIRO proteins comprise two GTPase domains surrounding a pair of EF-hands (Fig. 6B). To validate binding between RASSFs and MIRO we performed co-IPs of full-length MIRO1 (P13V) or MIRO2 (A13V) with numerous RASSFs (Fig. 6C). Both MIRO GTPases showed robust interaction with RASSF3 and RASSF4 and weak binding to RASSF5 and RASSF8. We further assayed whether these effectors could bind the MIRO-cGTPase domains. Co-IPs revealed none of the four RASSF effectors complex with MIRO1-cGTPase, whereas MIRO2-cGTPase associated strongly with RASSF3 and slightly less with RASSF4 and RASSF5 (Fig. 6D). To determine if binding between the RASSF RA domains and MIRO-nGTPase domains is direct we performed in vitro mixing assays using recombinantly purified proteins. MIRO GTPase domains were used directly following their purification from E. coli, or were exchanged with GDP, GTP, or GTP analogs (GMPPNP or GTPγS) to determine nucleotide dependency. The purified MIRO proteins were mixed with GST-RA domains from RASSF3 or RASSF5 or with GST-alone as a control. Following precipitation with glutathione beads and multiple washes, we observed a strong complex between RASSF3 and the nGTPase domains of MIRO1 and MIRO2 in a manner independent of loaded nucleotide (Fig. 6E). RASSF5 bound directly to purified KRAS in a nucleotide-dependent manner but did not strongly associate with MIRO1/2. The RA domain of RASSF3 did not bind to KRAS and the GST alone control did not precipitate any GTPases. Mixing assays with RASSF4 and RASSF8 showed the RA of RASSF8 also associates with MIRO (Fig. EV5B), but not RASSF4. This suggests that co-IP of RASSF4 and MIRO GTPases may not be reporting a direct interaction, may involve interfaces outside the RA domain, or that the RASSF4 RA is not properly folded following purification from E. coli. Nevertheless, these data designate RASSF3 as the first potential effector of the MIRO GTPases, which play important roles in mitochondrial homeostasis, trafficking, and PINK1/PARKIN-mediated mitophagy (Safiulina et al, 2019; López-Doménech et al, 2018).

There are no previous data describing RASSF3 subcellular localization, but interaction with mitochondrial-tethered MIRO GTPases should distribute this tumor suppressor protein to mitochondria. To investigate this, we transiently expressed VENUS-tagged MIRO1 or MIRO2 with mCherry-RASSF3. The RASSF3 protein alone showed a general cytosolic distribution, but its co-expression with MIRO1 or MIRO2 completely redistributed

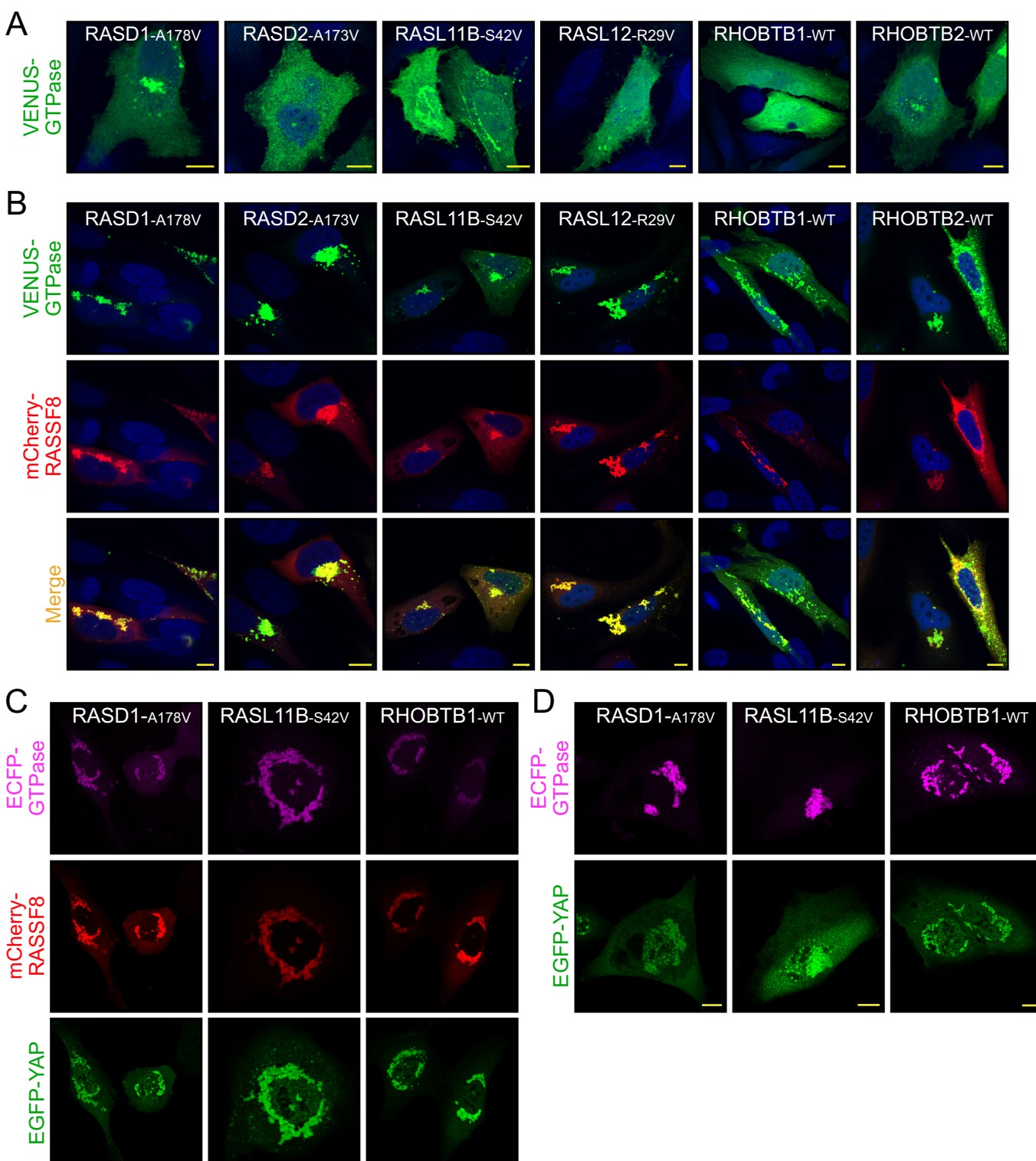

this effector to mitochondria (as detected by MitoTracker; Fig. 7A). Moreover, most cells expressing both RASSF3 and MIRO1/2 demonstrated a pattern of collapsed mitochondria around the perinuclear space, which was occasionally observed by expression of MIRO1/2 alone (Fig. EV5C,D). Quantitation of this phenotype revealed significant induction of mitochondrial network collapse

upon co-expression of MIRO GTPases and RASSF3 (Fig. 7B). Expression of RASSF3 alone did not significantly alter mitochondrial distribution. In addition to being tethered to mitochondria, a splicing variant of MIRO1 (MIRO1v4) has been reported associated with peroxisomes (Okumoto et al, 2018) (Fig. 7C). We also observed a peroxisomal localization of MIRO1v4 (confirmed by

**Figure 5. Multiple tumor suppressor-class RAS GTPases co-localize with RASSF8 and YAP in cells.**

(A) Representative images of mutationally activated VENUS-GTPases found as candidate partners for RASSF8 following their expression in HeLa cells. Nuclei are visualized with Hoechst (blue) and scale bars represent 10 µm. (B) Co-distribution of mCherry-RASSF8 and candidate VENUS-GTPases following co-expression in HeLa cells. Nuclei are in blue (Hoechst). Scale bars represent 10 µm. (C) Confocal microscopy images of U2OS cells co-transfected with EGFP-YAP, mCherry-RASSF8, and ECFP-GTPases as indicated. Scale bars represent 10 µm. (D) Representative images of U2OS cells co-expressing EGFP-YAP and the indicated ECFP-GTPases alone, in the absence of RASSF8. Scale bars represent 10 µm. All experiments were performed in three independent biological replicates. RASL11B-S42V and EGFP-YAP expressing cells were also transfected with mCherry alone and there was no bleed through to the red channel, as shown in Fig. EV4C. Source data are available online for this figure.

PEX14 staining), which no longer co-localized with mitochondria (Fig. 7D/E). Co-expression of RASSF3 with the MIRO1v4 variant resulted in collapsed clusters of both peroxisomes and mitochondria (Fig. 7D/E). This is highly reminiscent of the perinuclear clustering of peroxisomes and mitochondria induced by the overexpression of mitofusins (Huo et al, 2022; Alsayyah et al, 2024; Shai et al, 2018). Expression of RASSF3 or MIRO1v4 alone did not alter peroxisome distribution, and RASSF3 redistributed peroxisomal MIRO1v4 to mitochondrial clusters (Fig. 7E). These results corroborate RASSF3 as an effector of MIRO able to alter mitochondrial and/or peroxisome distribution, and further demonstrate a complex and versatile network of RASSF-GTPase interactions.

## Discussion

We have elucidated a map of BRAF and RASSF3/4/5/8 interactions with the RAS, RHO, and ARF subfamilies of small GTPases. Thirty-nine interactions were validated, most representing novel GTPase-effector complexes expected to regulate wide-ranging biological processes depending on context and the activation state of the identified G-protein. While H/K/NRAS remain the most studied proteins in the superfamily, alternative members should play fundamental roles in the biology of both normal and disease-state cells. The RAS subfamily itself is composed of 35 GTPases, some promoting growth while others have the hallmarks of tumor suppressors (Wennerberg et al, 2005; Bernal Astrain et al, 2022). Many are evolutionarily conserved yet are poorly characterized and lack known effector and regulatory proteins. For the effector proteins with RBD/RA domains (over 50 in the human proteome), RAF and PI3K remain the best characterized, attributed to their role in proliferation, survival, and oncogenesis (Smith, 2023). Classical RAS proteins also bind to effectors such as AFDN and RASSF5 (involved in cell adhesion and the Hippo pathway, respectively), indicating that GTPase-effector interactions can be complex and not necessarily highly specialized (Singh and Smith, 2020). Indeed, while the RBD of BRAF demonstrated high selectivity for H/K/NRAS, the RASSF domains exhibited much greater plasticity. These data suggest an array of RAS GTPases may signal through RASSF effectors in a manner dependent on co-expression, nucleotide cycling, and subcellular distribution in a biological context. This complexity has been poorly explored to-date, though previous work established that PI3K can be activated by 21 different GTPases of the RAS and RHO subfamilies (Yang et al, 2012). Moreover, an earlier study examined RASSF RA domain binding to seven purified RAS GTPases, revealing a multitude of potential low-affinity interactions (Adariani et al, 2021). Some of these were not detected by our approach and it remains unknown whether such weak interactions occur within a

cellular context, or how they might contend with higher affinity GTPase-effector complexes. Future use of large, systematic screens for GTPase-effector complexes should uncover an extensive array of novel and biologically relevant interactions that are context-dependent and difficult to detect with discovery-based approaches.

RASSF5 was the only family member to bind H/K/NRAS but was not highly selective, also interacting with the RAS-related proteins RAP2B/C, RRAS/2, MRAS, and RIT1/2. We determined that several of these interactions occur in a nucleotide-dependent manner which implicates RASSF5 as a canonical effector for these G-proteins. Further work is required to determine if all GTPase-effector complexes identified in our screen are governed by GDP-GTP cycling, or whether RASSF interactions with some partners may occur in a non-canonical fashion. We show these GTPases activate the Hippo pathway dependent on RASSF5, and previous work has elucidated important roles for both RIT1 and RAP2 in mediating Hippo activity and synergizing with nuclear YAP (Vichas et al, 2021; Meng et al, 2018). The Hippo pathway has recently garnered significant interest in cancer research and many Hippo proteins are now implicated in the tumorigenic process. Overexpression of YAP and TAZ is observed in many human cancers, including head and neck squamous cell carcinoma, cervical squamous cell carcinoma, prostate cancer, breast cancer, colorectal cancer, and non-small-lung cancer (Zanconato et al, 2016). Nuclear YAP/TAZ often correlates with poor prognosis and therapeutic resistance. The upstream Hippo pathway regulator NF2 and the kinases MST1/2 and LATS1/2 are bona fide tumor suppressor proteins (Wang et al, 2018). Moreover, multiple studies have discovered a functional link between Hippo and RAS signaling: loss of Hippo activity switches RAS activation from promoting differentiation to aggressive proliferation contingent on nuclear YAP, oncogenic RAS can induce YAP activation, and YAP can compensate for the loss of activated KRAS in pancreatic ductal adenocarcinoma (PDAC) (Muzumdar et al, 2017; Kapoor et al, 2014; Yan et al, 2021). YAP is also amplified and highly expressed in tumors that undergo spontaneous relapse and are devoid of KRAS-G12D, and inhibitors of the YAP co-regulator TEAD can overcome resistance to KRAS-G12C inhibitors (Kapoor et al, 2014; Hagenbeek et al, 2023). Unfortunately, there remain few data describing direct crosstalk between the RAS and Hippo pathways. Though RASSF5 is a direct interactor of both MST kinases and RAS GTPases, its role in regulating Hippo signaling is poorly understood. Mechanistically, RASSF proteins inhibit MST kinase activity in vitro (Ni et al, 2013). This suggests that RASSF5 recruitment by activated GTPases in cells may release bound MST to enable kinase activation. However, our observation that RIT2 engagement with RASSF5 does not promote cytoplasmic YAP suggests something more complex is required. This could be a component on the plasma membrane, where all other candidate GTPase partners of RASSF5 identified here reside. A more complex MST activation

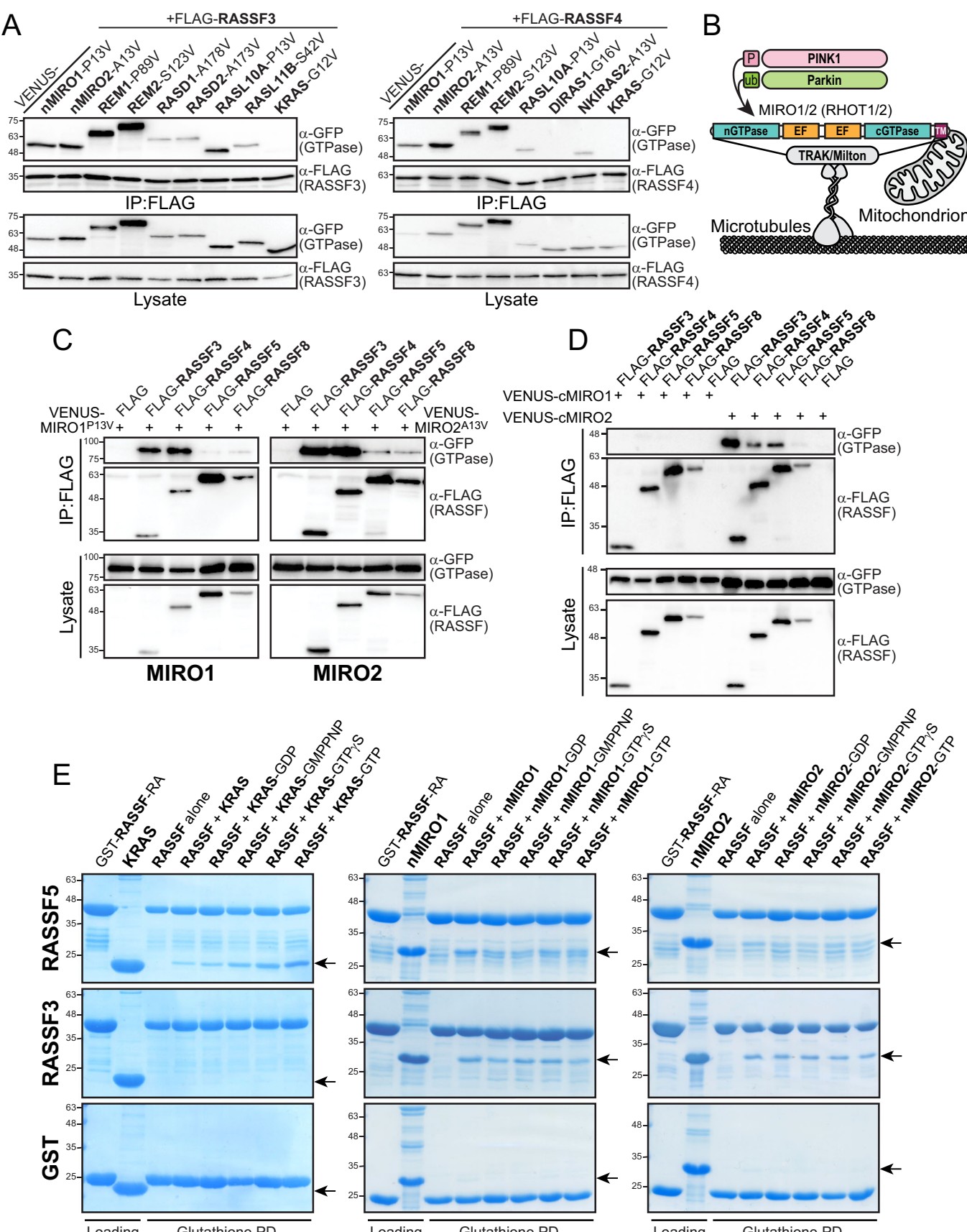

Figure 6. RASSF3 interacts with the mitochondrial GTPases MIRO1 and MIRO2.

(A) Co-IP of candidate GTPases identified as hits for RASSF3 and RASSF4. FLAG-tagged RASSF3 (left) or RASSF4 (right) were co-transfected with mutationally activated, VENUS-GTPases in HEK 293T cells and lysates were subjected to anti-FLAG IP. (B) Schematic of modular domains in MIRO1/2 and the role of MIRO GTPases in mitochondrial function. MIROs are tethered to the mitochondrial outer membrane and bridge the organelles with microtubule-binding proteins TRAK/Milton to regulate transport. MIROs are also major ubiquitination targets of PINK1/Parkin in mitophagy. (C) Anti-FLAG co-IP of full-length mutationally activated variants VENUS-MIRO1-P13V or VENUS-MIRO2-A13V with full-length FLAG-RASSF3, -RASSF4, -RASSF5, or -RASSF8 following their co-expression in HEK 293T cells. (D) Co-IP of VENUS-tagged cGTPase domains of MIRO1 and MIRO2 with full-length FLAG-RASSF3, -RASSF4, -RASSF5, or -RASSF8 following co-expression in HEK 293T cells. (E) In vitro mixing assays using recombinantly purified GST-tagged RA domains of RASSF5 (top), RASSF3 (middle), or GST alone (bottom) to precipitate the small GTPase KRAS (left), nGTPase domain of MIRO1 (nMIRO1, middle), or nGTPase domain of MIRO2 (nMIRO2, right) on glutathione beads. The arrow indicates where the precipitated GTPases should appear if bound. GTPases were pre-loaded with the nucleotides indicated at the top. PD stands for pull down. Source data are available online for this figure.

mechanism requiring its translocation to specific subcellular locales (e.g., PM) would explain why the simple loss of *RASSF* expression via promoter hypermethylation, frequently observed in cancer, is not alone able to activate Hippo. More data is required to understand how distinct GTPases gatekeep Hippo activity through RASSFs, and how this could be exploited for cancer therapy. Further investigation is also required to understand how RASSF5 interactions with growth-promoting RAS subfamily GTPases impact proliferation in normal cells, and how these complexes impact MAPK signaling. We postulate these signaling complexes will negatively regulate proliferation, operating as checkpoints for multiple cell states that crosstalk to the Hippo pathway. This is the likely reason why immortalized cells in culture have low levels of endogenous RASSF proteins.

There have been comparatively few studies of RASSF3, RASSF4, and RASSF8, though it is clear their expression is downregulated in numerous human cancers (Richter et al, 2009). Our screen uncovered the first small GTPase partners for these effectors, predominantly in the tumor suppressor-class RAS subfamily. In cultured cells, RASSF8 undergoes LLPS and is distributed in membraneless organelles. Biomolecular condensation has emerged as an important mechanism of subcellular organization and spatiotemporal control of biochemical reactions, including for many components of the Hippo pathway (Su et al, 2021). Angiomotins, KIBRA and SLMAP1 form condensates in response to cell–cell contact, and LLPS of the downstream effectors YAP and TAZ regulates transcription (Wang et al, 2022; Cai et al, 2019; Lu et al, 2020). Network data implicate RASSF8 in Hippo signaling despite the lack of a SARAH motif in the N-RASSFs, and we confirmed that RASSF8 condensates also contain YAP. This suggests a universal function for RASSF proteins in regulating the Hippo pathway downstream of activated GTPases. The RASD, RASL, RGK, NKIRAS, and RHO GTPases verified as RASSF8 binders change the dynamics of RASSF8 condensates and drives a liquid-to-solid phase transition of RASSF8. Many protein condensates associated with neurological disorders, such as α-synuclein, FUS, and TDP-43 undergo a similar transition from the liquid-to-solid state (Patel et al, 2015; Carey and Guo, 2022; Ray et al, 2020). The GTPase partners of RASSF8 could all play roles in regulating RASSF8-YAP condensation and, consequently, YAP transcriptional activity. As there are few data on the regulatory or activation status of these small G-proteins, elucidating their role in regulating YAP activity through RASSF8 should begin with characterizations of their nucleotide cycling and identification of regulatory proteins and/or environmental contexts that preclude binding to the RASSF effectors. This will offer insight into novel pathways linking cell–cell adherence to YAP and cell proliferation.

RASSF3 shares 48% amino acid sequence identity with RASSF5 but displayed a markedly distinct interaction profile with GTPases.

These were tumor suppressor-class RAS proteins with anti-proliferative and pro-apoptotic functions, along with the MIRO-nGTPase domains. We found both MIRO1 and MIRO2 complex with RASSF3 in vitro, confirming direct interaction, though binding appeared independent of GTP-loading. It has been challenging to biochemically characterize MIRO nucleotide cycling with some suggesting it is constitutively GTP-bound or does not behave like a "molecular switch" (Smith et al, 2020). We also had difficulty working with purified MIRO nGTPases, as they were prone to aggregation and poorly behaved in solution. Thus, we cannot confirm differential nucleotide loading in vitro and it may emerge MIRO interactions with RASSF3 in cells are indeed GTP-dependent or may occur in an otherwise non-canonical fashion. Nevertheless, RASSF3 is the first potential effector of MIRO1/2 and was recruited to mitochondria when expressed with these GTPases. Co-expression of RASSF3 and MIRO resulted in a collapsed mitochondrial network, suggesting disruption of MIRO connection to microtubules. Similar perinuclear mitochondrial clustering was previously observed upon MIRO or Mfn1/2 overexpression, and with Charcot-Marie-Tooth syndrome Type 2A (CMT2A)-associated Mfn variants and disease-driving PARKIN mutants (Huo et al, 2022; Sloat and Hoppins, 2023; Huang et al, 2007; Lee et al, 2010). RASSF3 also promoted the clustering of peroxisomes in this region and led to the redistribution of peroxisomal MIRO1v4 with mitochondrial clusters. Further investigation of how RASSF3 alters MIRO-microtubule interactions will shed light on how this GTPase-effector complex regulates the trafficking of these organelles, with significant implications for neurodegenerative diseases and other human disorders.

Overall, our findings support a highly complex network of effector-GTPase interactions that enable cells to evoke intracellular signaling pathways downstream of diverse stimuli. The RASSF family as a whole should now be considered Hippo regulators that are controlled by a disparate set of RAS GTPases. Contextual data elucidating how these G-proteins are activated and their impact on YAP activity will further clarify upstream signals feeding into the Hippo-driven regulation of cell proliferation.

## Methods

### Constructs and antibodies

Gateway Entry vectors encoding 83 mutationally activated human small GTPases of the RAS, RHO, and ARF subfamilies were a kind gift from Dr. Jean-François Côté (IRCM, Montreal) (Patel et al, 2011). GTPases were shuttled into a mammalian expression

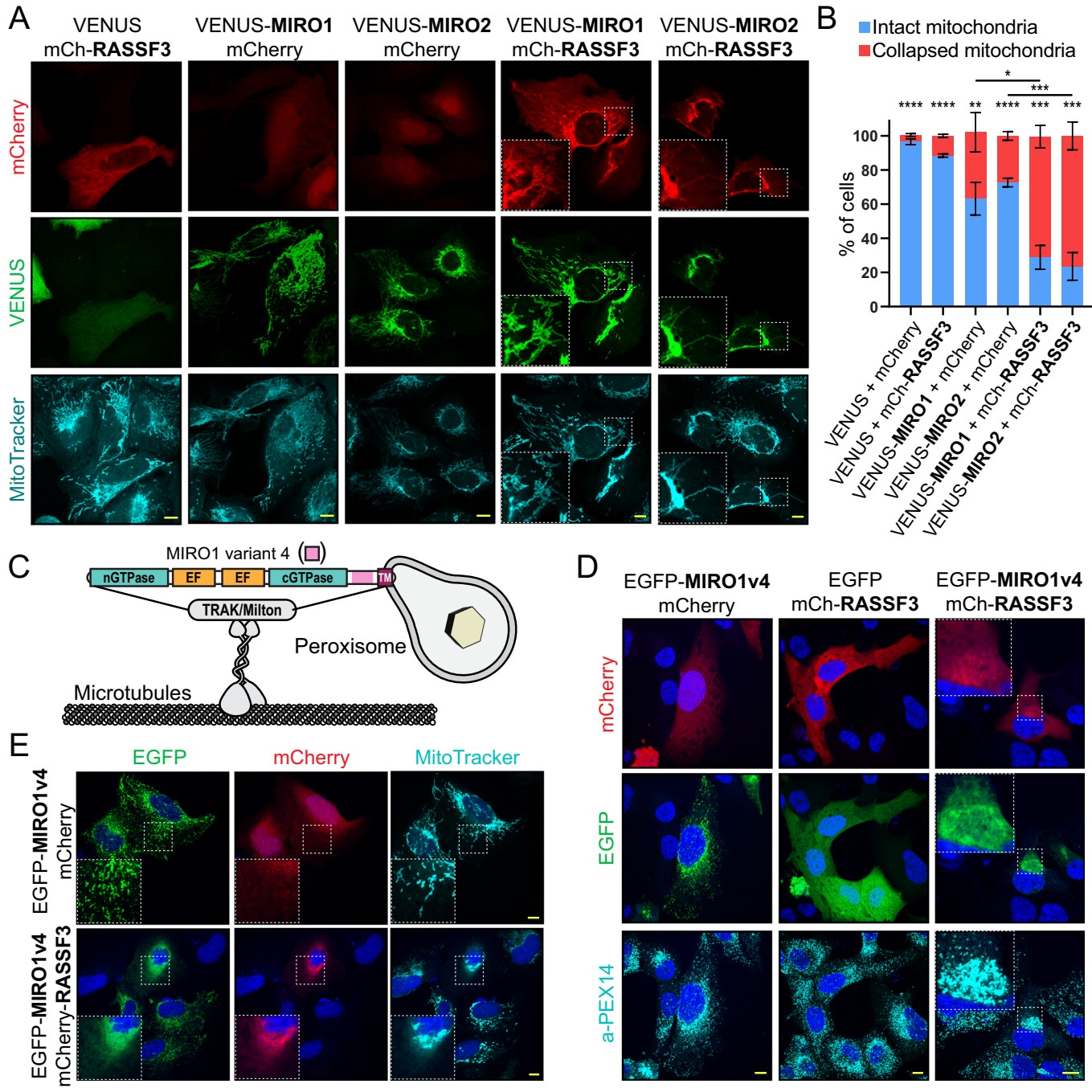

**Figure 7. RASSF3 is recruited to mitochondria when co-expressed with MIRO GTPases.**

(A) Representative images show co-localization of VENUS-tagged MIRO1 and MIRO2 with mCherry-RASSF3 at mitochondria in U2OS cells. Mitochondria were stained using MitoTracker. Scale bars represent 10 μm. (B) Quantification of collapsed and intact mitochondrial networks in individual U2OS cells expressing RASSF3 or MIRO GTPases alone, or in combination. Data represent mean ± SD, $n = 3$ biological replicates, >75 cells for each condition, multiple unpaired $t$-test, **** represents $P < 0.0001$, ***$P < 0.001$, **$P < 0.01$, and *$P < 0.05$. (C) Schematic representation of the MIRO1 variant 4 (MIRO1v4) previously shown to associate with peroxisomes and believed to bridge these organelles with microtubules. (D) Immunostaining of endogenous PEX14 (peroxisome marker) in U2OS cells co-expressing EGFP-MIRO1v4 and mCherry-RASSF3. Nuclei are visualized with Hoechst (blue). Scale bars represent 10 μm. (E) Confocal images of U2OS cells expressing EGFP-MIRO1v4 alone or with mCherry-RASSF3. Mitochondria were stained with MitoTracker (cyan) and nuclei were stained with Hoechst (blue). Scale bars represent 10 μm. Data represent $n = 3$ independent experiments. Source data are available online for this figure.

Gateway destination vector (with N-terminal VENUS-tag). cDNAs encoding human RASSF3/4/5/8 were provided by Dr. Anne-Claude Gingras (LTRI, Toronto). These were cloned into a Gateway Entry vector and recombined into mammalian expression Gateway destination vectors with either N-terminal VENUS, mCherry or FLAG tags. RBD/RA domains were cloned into pGEX-4T2 (Amersham Pharmacia Biotech), a bacterial expression vector with an N-terminal Glutathione S-transferase (GST) tag, using domain boundaries described previously (Dhanaraman et al, 2020). For in vitro mixing assays, the G-domain of KRAS4B (1–185 amino acids) was cloned into the pET28 bacterial expression system with an N-terminal 6x-Histidine (His) tag. Human MIRO1-nGTPase (1–180) and MIRO2-nGTPase (1–180) were cloned in the pET28a bacterial expression system with an N-terminal 6xHis tag. EGFP-MIRO1v4 from Dr. Pietro De Camilli was obtained from Addgene (#174111). Human MIRO1-cGTPase (410-585) and MIRO2-cGTPase (410–605) were cloned in a Gateway Entry vector and recombined into mammalian expression Gateway destination vectors with N-terminal VENUS-tag. The expression construct for EGFP-tagged G3BP1 was a gift from Dr. Jean-Philippe Gratton (Université de Montréal, Montréal) and EGFP-YAP from Dr. Sylvain Meloche (IRIC, Montréal). Antibodies used include anti-EEA1 (BD Transduction Laboratories, 610457; IF: 1:100), anti-Golgin-97 (Thermo Fisher, A-21270; IF: 1:100), anti-GM130 (Abcam, ab30637; IF: 1:100), anti-YAP (63.7; Santa Cruz Biotechnology, sc-101199; IF: 1:100; WB: 1:500), anti-PEX14 (Sigma, HPA049231; IF: 1:150), anti-RASSF5 (Abcam, ab33292; WB: 1:1000), anti-RASSF3 (Sigma-Aldrich, HPA038469; WB: 1:1000), anti-RASSF8 (Sigma-Aldrich, HPA038164; WB: 1:1000), anti-GFP (Abcam, ab290; WB: 1:5000), anti-FLAG M2 (Sigma, F3165; IF: 1:100; WB: 1:1000), anti-mouse Tx-Red (Sigma, SAB3701076; IF: 1:100), anti-rabbit Alexa Fluor 488 (Life Technologies, A-27034; IF: 1:200), HRP-conjugated anti-rabbit (Cedarlane, NA934; WB: 1:10,000) and HRP-conjugated anti-mouse IgG (Fisher, 45-000-679; WB: 1:10,000).

## Purification of recombinant proteins and nucleotide exchange

GST-tagged RBD/RA domains and the 6xHis-KRAS G-domain were expressed and purified as described previously (Dhanaraman et al, 2020). Briefly, recombinant proteins were expressed in *Escherichia coli* (BL21-DE3-CodonPlus) cells grown in LB medium at 37 °C. *E.coli* cells expressing GST-RBD and 6xHIS-KRAS were induced with 250 μM isopropyl-β-D-thiogalactopyranoside (IPTG), and grown overnight at 16 °C. Cells were harvested and lysed in 20 mM Tris-HCl (pH 7.5), 150 mM NaCl, 5 mM MgCl$_2$, 10% glycerol (v/v), 0.4% NP-40 (v/v), protease inhibitors (Roche), and with either 1 mM dithiothreitol (DTT), or 10 mM β-mercaptoethanol and sonicated. Lysates were cleared by centrifugation at 15,000 rpm for 30 min. GST-RBD and 6xHis-KRAS were bound to glutathione (Amersham) or Ni$^{2+}$-nitrilotriacetic acid [Ni-NTA (Qiagen)] resins, respectively, at 4 °C for 1–2 h. Bound GST-RBD and 6xHis-KRAS were eluted with 30 mM reduced glutathione or 250 μM imidazole, respectively, and further purified using size exclusion chromatography (20 mM Tris-HCl (pH 7.5), 150 mM NaCl, 5 mM MgCl$_2$, and 2 mM DTT) using a Superdex 75 16/600 (GE Healthcare) column. Purification of 6xHis-tagged MIRO nGTPase was adapted from a previous study (Smith et al,

2020). Briefly, *E. coli* cells expressing 6xHis-MIRO1/2-nGTPase domains were induced with 125 mM IPTG and then grown overnight at 16 °C. Cells were resuspended in 50 mM HEPES-HCl (pH 8.0), 500 mM NaCl, 5 mM MgCl$_2$, 0.5 mM TCEP, 5% sucrose (w/v), and 0.05% Tween20 (v/v), followed by sonication and clarification by centrifugation at 18,000 rpm for 30 min. The protein was bound on Ni-NTA resins at 4 °C for 2 h and eluted using lysis buffer supplemented with 250 mM imidazole. Proteins were further purified (50 mM HEPES-HCl (pH 8.0), 500 mM NaCl, 5 mM MgCl$_2$, 0.5 mM TCEP, 5% sucrose) using a 16/600 S75 size exclusion column. For nucleotide exchange, GTPases were incubated with a tenfold molar excess (nucleotide:protein) of GMPPNP, GTPγS, or GTP (Sigma-Aldrich) and 10 mM EDTA at 37 °C for 10 min. They were immediately put on ice and 20 mM MgCl$_2$ was added. Proteins were then dialyzed at 4 °C or run through an S75 size exclusion column.

## Cell culture and microscopy

HEK 293T (American Type Culture Collection (ATCC) CRL-3216), HeLa (ATCC CCL-2), and U2OS (ATCC HTB-96) cell lines were maintained in Dulbeeco's Modified Eagle Medium (DMEM) supplemented with a 20% fetal bovine serum (FBS)/80% new calf serum (NCS) mix for HEK 293T and 10% FBS for other cell lines. All cultured lines underwent bi-weekly testing for mycoplasma contamination. For imaging, HeLa or U2OS cells were seeded on six-well plates containing coverslips and transfected using jet-PRIME or Lipofectamine 3000, respectively. Forty-eight hours after transfection, cells were washed with phosphate-buffered saline (PBS) and fixed with 4% paraformaldehyde. For mitochondrial staining, live cells were incubated with MitoTracker™ Deep Red FM dye (Invitrogen by Thermo Fisher Scientific) for 30 min and then fixed with 4% formaldehyde at 37 °C for 15 min. HeLa cells were permeabilized using PBS with 0.05% Tween and blocked with 4% FBS. U2OS cells were permeabilized using 0.1% Triton X-100 for 5 min and then blocked with 2% BSA for 1 h. Cells were further incubated for 1 h with primary antibody at 37 °C (or 2 h at RT for anti-YAP 63.7) and washed three times with blocking buffer, followed by incubation with secondary antibody and Hoechst. After the final washes with blocking buffer, coverslips were treated with 70% and then 95% ethanol. Coverslips were air-dried before being mounted on glass slides. Images were obtained using a Zeiss LSM880 confocal microscope and processed using ZEN software. FRAP assays were conducted using live U2OS cells in a 37 °C, 5% CO$_2$ chamber on a Zeiss LSM880 confocal microscope. A defined region was photobleached by the equipped laser at 100% power intensity, followed by time-lapse live imaging with a 10 s interval to record fluorescence recovery.

## Recombinant protein pull-downs of GTPases in mammalian cell lysate

For the RBD/RA-GTPase screen, HEK 293T cells were seeded on 24-well plates (at $0.2 \times 10^6$ cell confluency) and transiently transfected with 500 ng of DNA using polyethylenimine (PEI). Forty-eight hours after transfection, cells were harvested and lysed (20 mM Tris-HCl (pH 7.5), 150 mM NaCl, 5 mM MgCl$_2$, 10% glycerol, 1% Triton X-100, 1% NP-40, 1 mM DTT, and protease inhibitor P3840) for 10 min. The lysate was clarified by

centrifugation at 15,000 rpm for 15 min. The supernatant was incubated with 2 μM of recombinantly purified GST-tagged RBD/RA domains of RASSF3/4/5/8, BRAF, or GST-alone and glutathione beads for 30 min. The mixture of beads with GTPases in lysate and GST-tagged proteins were transferred to a Bio-Spin column and kept on a vacuum manifold, wherein lysis buffer was passed three times for washing. Columns were put onto microfuge tubes to elute the sample with bound proteins by reconstituting beads in an SDS-loading buffer, followed by heating and centrifugation. For Western blotting, protein samples were separated using SDS-polyacrylamide gel electrophoresis (SDS-PAGE) and transferred to nitrocellulose membranes. Membranes were blocked with Tris-buffered saline plus 0.1% Tween (TBST) containing 5% skim milk (or 3% bovine serum albumin (BSA) for phospho-protein antibodies), followed by overnight incubation with primary antibody, which was later detected using HRP-conjugated anti-rabbit or anti-mouse immunoglobulin (Ig) followed by treatment with enhanced chemiluminescence reagent (Bio-Rad). The signal was detected using a Bio-Rad ChemiDoc imaging system and analyzed with ImageLab software. Densitometry data were collected by quantifying band intensities for each interaction observed for GTPase with GST-tagged proteins using ImageLab. The intensity from a control RASSF5-KRAS-G12V interaction on each blot was used for normalization between experiments. The intensities from precipitation of VENUS-tagged GTPases by GST-alone were subtracted from intensities observed for GTPase binding with GST-tagged RBD/RA domains to remove non-specific binding with the GST-tag. Following at least two repeats of every pull-down experiment, data were compiled to create a heatmap in Excel using a three-color gradient representing strong, moderate, and no binding.

### Co-precipitation assays

For co-immunoprecipitations, HEK 293T cells seeded on six-well plates were co-transfected with FLAG-tagged RASSFs and VENUS-tagged GTPases (PEI). After 48 h, harvested cells were lysed (20 mM Tris-HCl (pH 7.5), 150 mM NaCl, 5 mM MgCl₂, 10% glycerol, 1% Triton X-100, 1 mM DTT, and protease inhibitor P3840) and kept on ice for 10 min. The supernatant was collected after centrifugation at 15,000 rpm for 15 min. Lysates were incubated for 1 h with prewashed Protein G Sepharose bound with the immunoprecipitating antibody (anti-FLAG). After incubation, beads were washed three times with lysis buffer followed by reconstitution in SDS-loading buffer. Interactions were analyzed using Western blotting with anti-GFP. For GST mixing experiments, 10 μM of recombinantly purified GST-tagged RASSF RA domains bound to glutathione beads, or GST alone control, were incubated for 2 h with 50 μM purified 6xHis-GTPase, either exchanged with GDP, GMPPNP, GTPγS, GTP, or unexchanged. Beads were washed three times by centrifugation using lysis buffer (see pull-down assay) and were then reconstituted in SDS-sample buffer and separated by SDS-PAGE.

### Quantitation of puncta or collapsed mitochondrial networks in cells

Quantification of cellular phenotypes was performed using Zeiss ZEN software on images taken on an LSM880 confocal microscope

and were not conducted with blinding. Statistical analyses were performed using GraphPad Prism. All experiments were performed in triplicate and data were presented as mean ± standard deviation (SD). Multiple unpaired *t*-tests were performed to indicate statistical significance.

## Data availability

This study includes no data deposited in external repositories.

The source data of this paper are collected in the following database record: biostudies:S-SCDT-10_1038-S44319-024-00203-9.

## Peer review information

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

## Acknowledgements

This work was supported by grants (to M.J.S.) from the Canadian Institutes for Health Research (CIHR), the National Science and Engineering Council of Canada (NSERC), and the Cancer Research Society (CRS). M.J.S. holds a Canada Research Chair in Cancer Signaling and Structural Biology. SS and GBA were supported by scholarships from the Fonds de Recherche du Québec-Santé (FRQS), and SS by a scholarship from the Fonds de Recherche du Québec-Nature et technologies (FRQNT). We would like to acknowledge Christian Charbonneau at the Bio-Imaging Core Facility (IRIC) for guidance with confocal imaging.

## Author contributions

**Swati Singh**: Conceptualization; Data curation; Formal analysis; Validation; Investigation; Visualization; Methodology; Writing—original draft; Writing—review and editing. **Gabriela Bernal Astrain**: Data curation; Formal analysis; Investigation. **Ana Maria Hincapie**: Investigation. **Marilyn Goudreault**: Resources; Formal analysis; Investigation. **Matthew J Smith**: Conceptualization; Resources; Data curation; Formal analysis; Supervision; Funding acquisition; Visualization; Methodology; Writing—original draft; Project administration; Writing—review and editing.

Source data underlying figure panels in this paper may have individual authorship assigned. Where available, figure panel/source data authorship is listed in the following database record: biostudies:S-SCDT-10_1038-S44319-024-00203-9.

## Disclosure and competing interests statement

The authors declare no competing interests.

# Expanded View Figures

**Figure EV1. Mapping BRAF and RASSF interactions with the RHO and ARF subfamilies of small GTPases.**

(A) Heatmap of BRAF RBD and RASSF3/4/5/8 RA binding to activated GTPases of the RHO subfamily. Strong interactions are in red, moderate in yellow and no interaction in black. (B) Heatmap of BRAF RBD and RASSF3/4/5/8 RA domains binding to activated GTPases of the ARF subfamily. ARL17, ARL9, and ARL16 were not included in the screen. Strong interactions are in red, moderate in yellow and no interaction in black. (C) Recombinantly purified GST-RASSF5 RA complexes with 10 distinct RAS subfamily GTPases following expression of VENUS-tagged, mutationally activated GTPases in HEK 293 T cells. RHEB served as a negative control. (D) Co-immunoprecipitation of ten candidate RAS subfamily GTPases following co-expression of the VENUS-tagged, mutationally activated variants with FLAG-RASSF5.

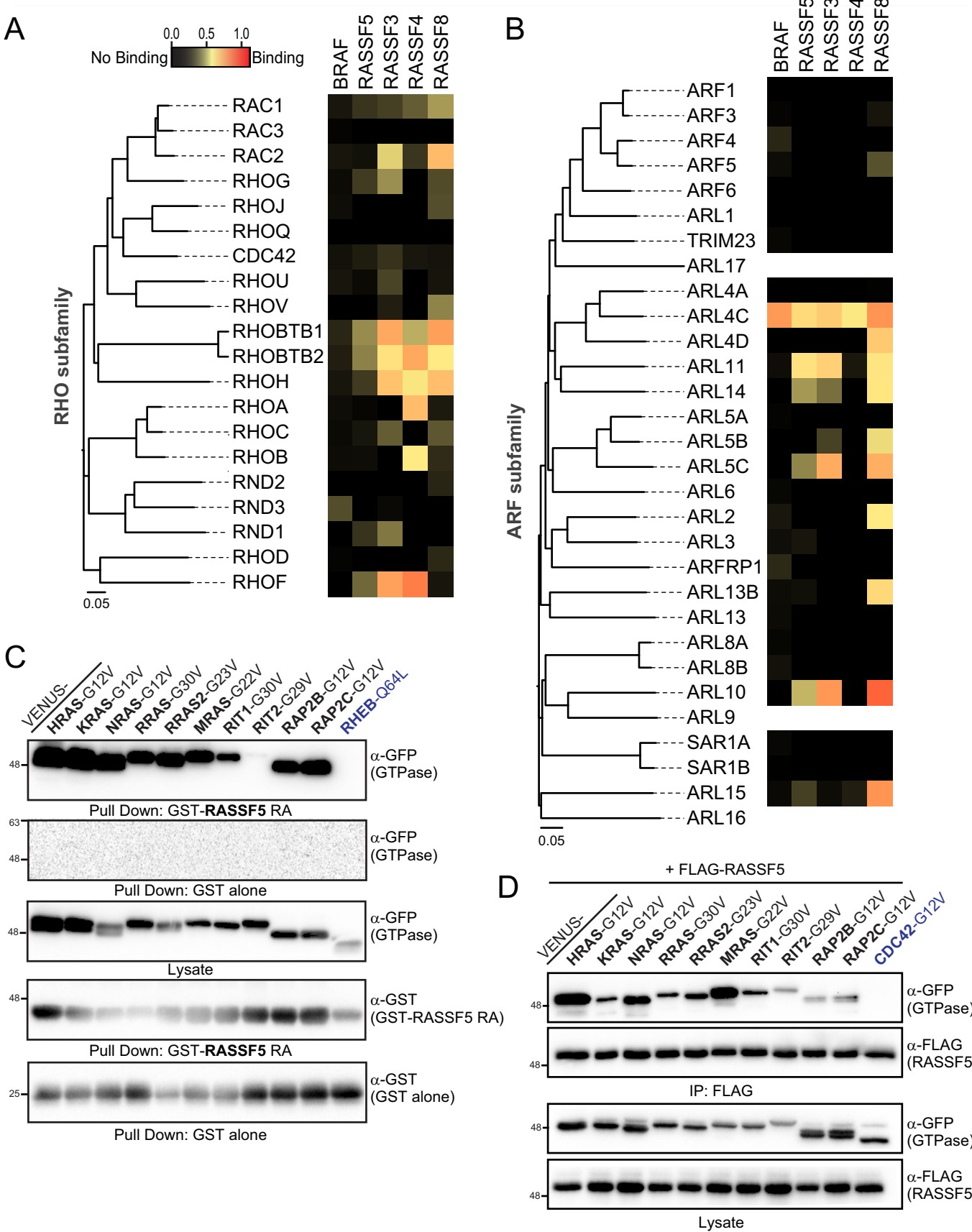

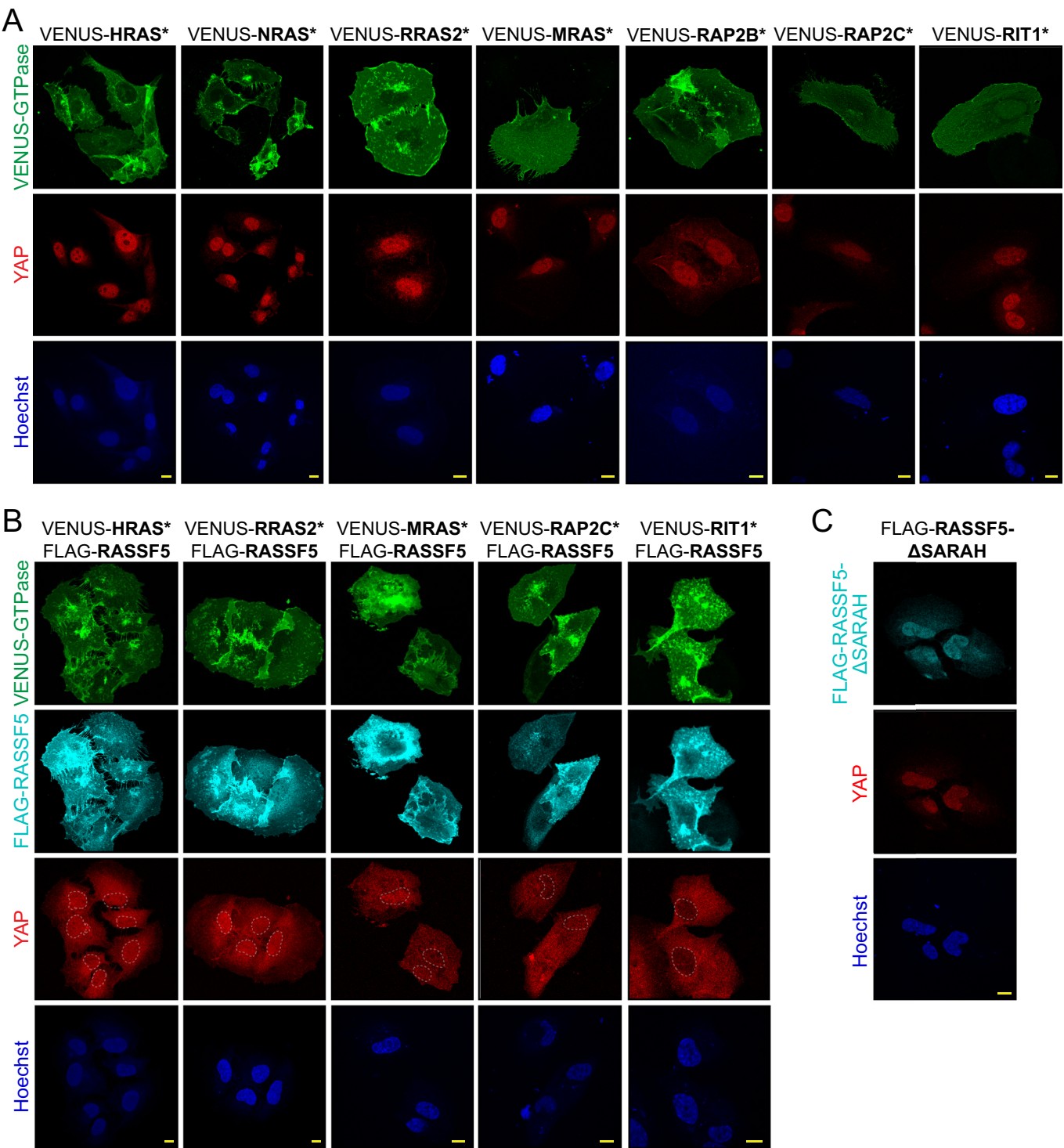

**Figure EV2.   The RAS subfamily GTPase partners of RASSF5 activate the Hippo pathway in a RASSF5-dependent manner.**

(A) Immunofluorescence images of endogenous YAP (red) in sparsely confluent U2OS cells transiently expressing mutationally activated (*) VENUS-tagged HRAS-G12V, NRAS-G12V, RRAS2-G23V, MRAS-G22V, RAP2B-G12V, RAP2C-G12V, or RIT1-G30V. Nuclei were visualized with Hoechst staining (blue). Scale bars represent 10 µm. (B) Immunostaining of endogenous YAP (red) in sparsely confluent U2OS cells co-transfected with FLAG-RASSF5 (cyan) and mutationally activated (*) GTPases (HRAS-G12V, RRAS2-G23V, MRAS-G22V, RAP2C-G12V or RIT1-G30V). Hoechst staining (blue) was used to visualize nuclei, and scale bars represent 10 µm. (C) Immunostaining for YAP (red) in U2OS cells expressing FLAG-RASSF5-ΔSARAH (cyan). Scale bars represent 10 µm.

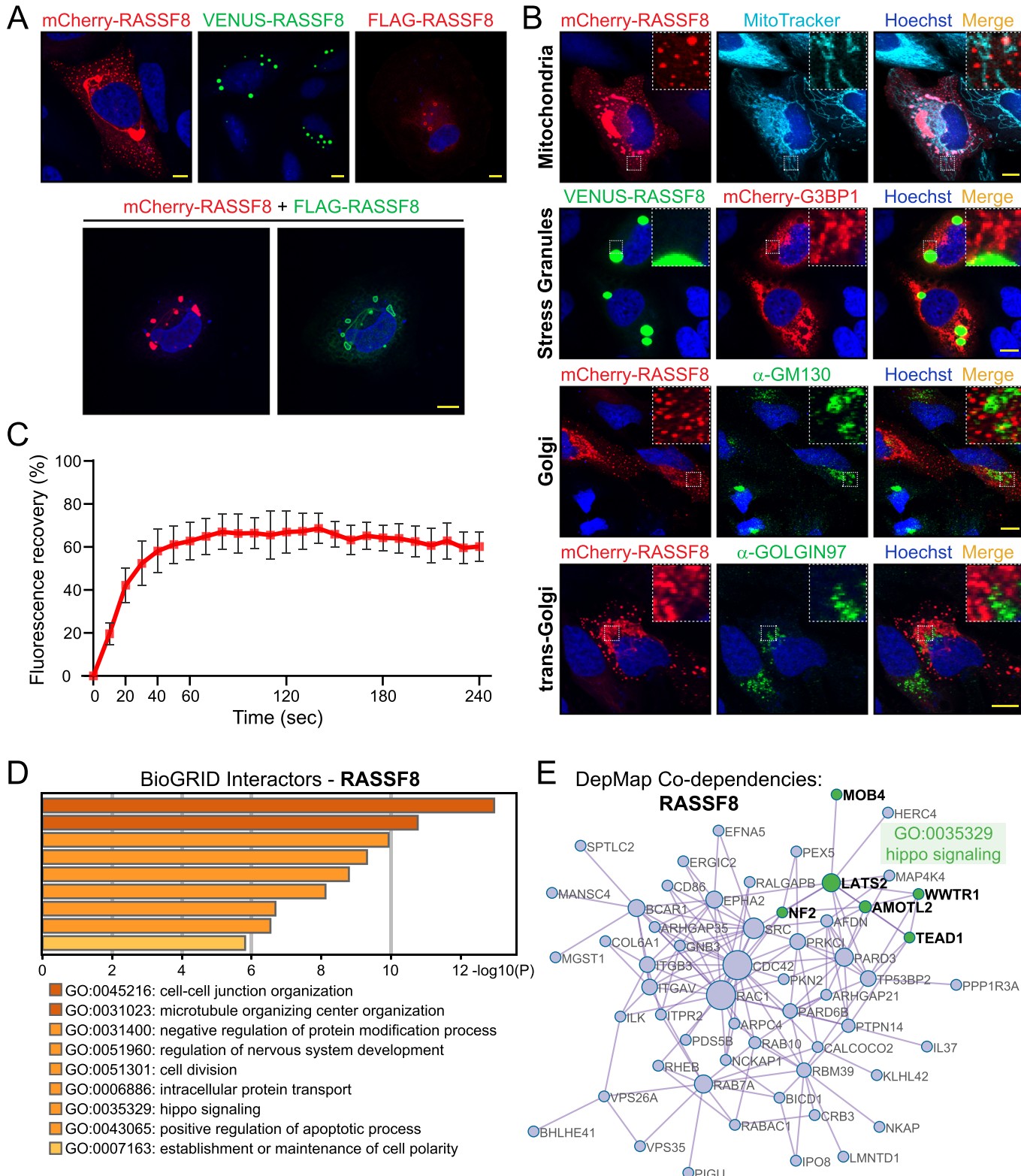

**Figure EV3. Characterization of RASSF8 puncta in cells.**

(A) Representative confocal microscopy images of U2OS cells expressing mCherry-RASSF8, VENUS-RASSF8, or FLAG-RASSF8 (top row). Also shown are cells co-expressing mCherry-RASSF8 with FLAG-RASSF followed by immunostaining with anti-FLAG (bottom row). Nuclei were visualized with Hoechst (blue). Scale bars represent 10 μm. (B) Fluorescence images of U2OS cells transiently expressing mCherry-RASSF8. These were stained markers for mitochondria (MitoTracker), golgi (anti-GM130), or trans-golgi (GOLGIN97). Alternatively, cells expressing VENUS-RASSF8 were co-transfected with a marker for stress granules (mCherry-G3BP1). Nuclei are in blue. Scale bars represent 10 μm. (C) Quantification of FRAP recovery as a fraction of the initial fluorescence intensity of mCherry-RASSF8. Data presented as mean ± SD, $n = 10$ biological replicates. (D) Gene Ontology (GO) analysis of all RASSF8 interactors found in the BioGRID database, conducted using Metascape. Enriched GO terms are listed at bottom. Bars are colored by increasing $P$ values as determined by Metascape (using a hypergeometric test and Benjamini–Hochberg $p$ value correction algorithm). (E) Network analysis of RASSF8 genetic interactors as identified by the CRISPR-based cancer dependency map (DepMap). Enriched networks were identified by Metascape, and include multiple proteins related to Hippo signaling.

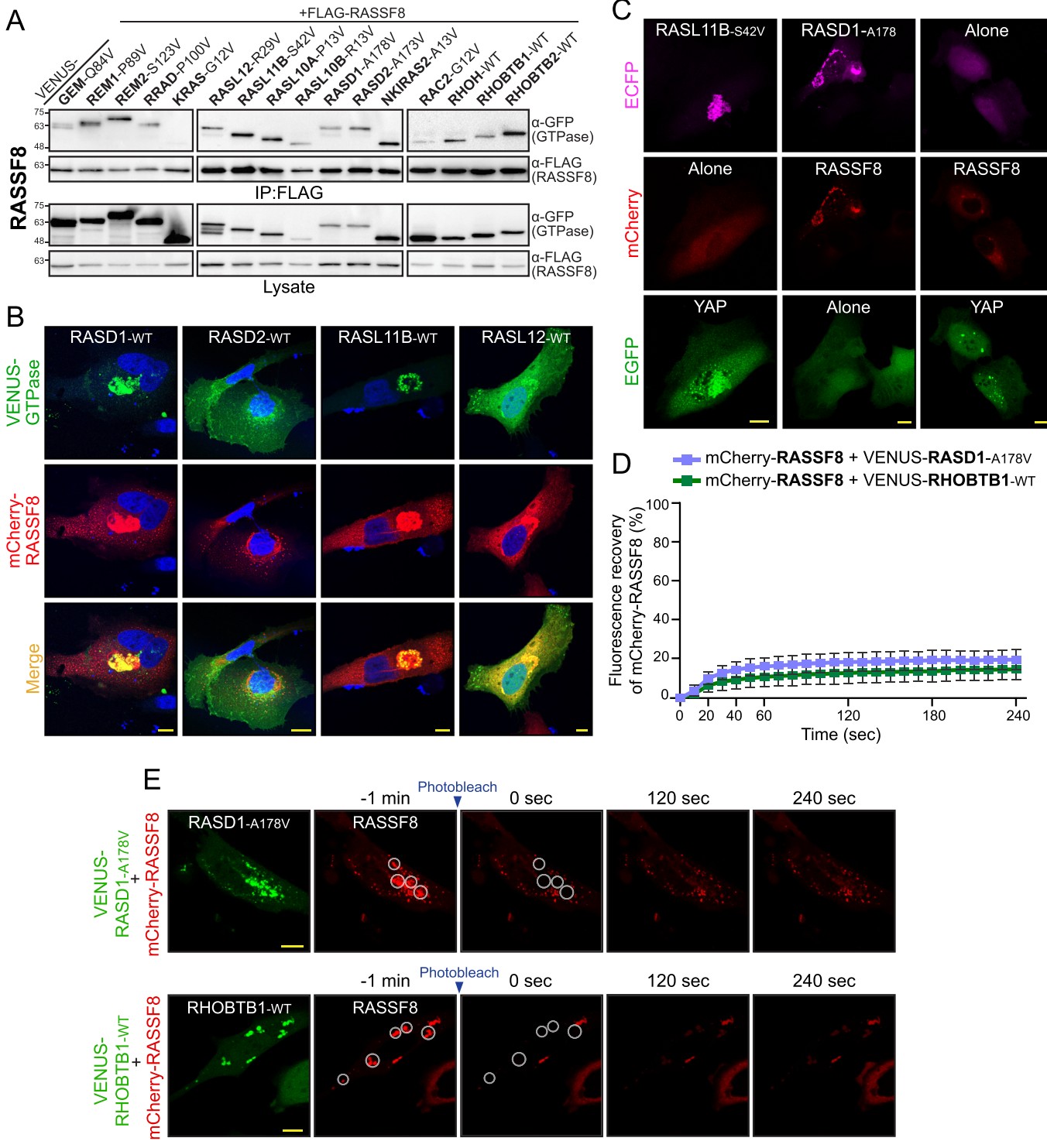

**Figure EV4.   Validation of RASSF8 interactions with RAS subfamily GTPases.**

(**A**) Co-IP of 15 candidate RAS and RHO subfamily GTPases following co-expression of the VENUS-tagged, mutationally activated variants with FLAG-RASSF8 in HEK 293T cells. KRAS-G12V does not interact with RASSF8 and served as a negative control. RHOH and the RHOBTB proteins are pseudoGTPases and were assayed as wild-type. (**B**) Co-distribution of mCherry-RASSF8 and four wild-type candidate VENUS-GTPases following co-expression in HeLa cells. Nuclei are in blue (Hoechst). Scale bars represent 10 μm. (**C**) Controls for confocal images in U2OS cells co-expressing EGFP-YAP, ECFP-GTPases, and mCherry-RASSF8. No bleed through was observed in corresponding channels when EGFP, ECFP, or mCherry tags were expressed alone. Scale bars represent 10 μm. (**D**) Quantification of FRAP recovery as a fraction of the initial fluorescence intensity of mCherry-RASSF8 co-expressed with either RASD1-A178V or RHOBTB1-WT. Data presented as mean ± SD, $n = 10$ biological replicates. (**E**) Representative images from time-lapse live-cell imaging of U2OS cells expressing mCherry-RASSF8 with VENUS-tagged RASD1-A178V or RHOBTB1-WT. Encircled mCherry-RASSF8 puncta were photobleached and exhibited notably reduced fluorescence recovery. Time is indicated above panes and scale bar represents 10 μm.

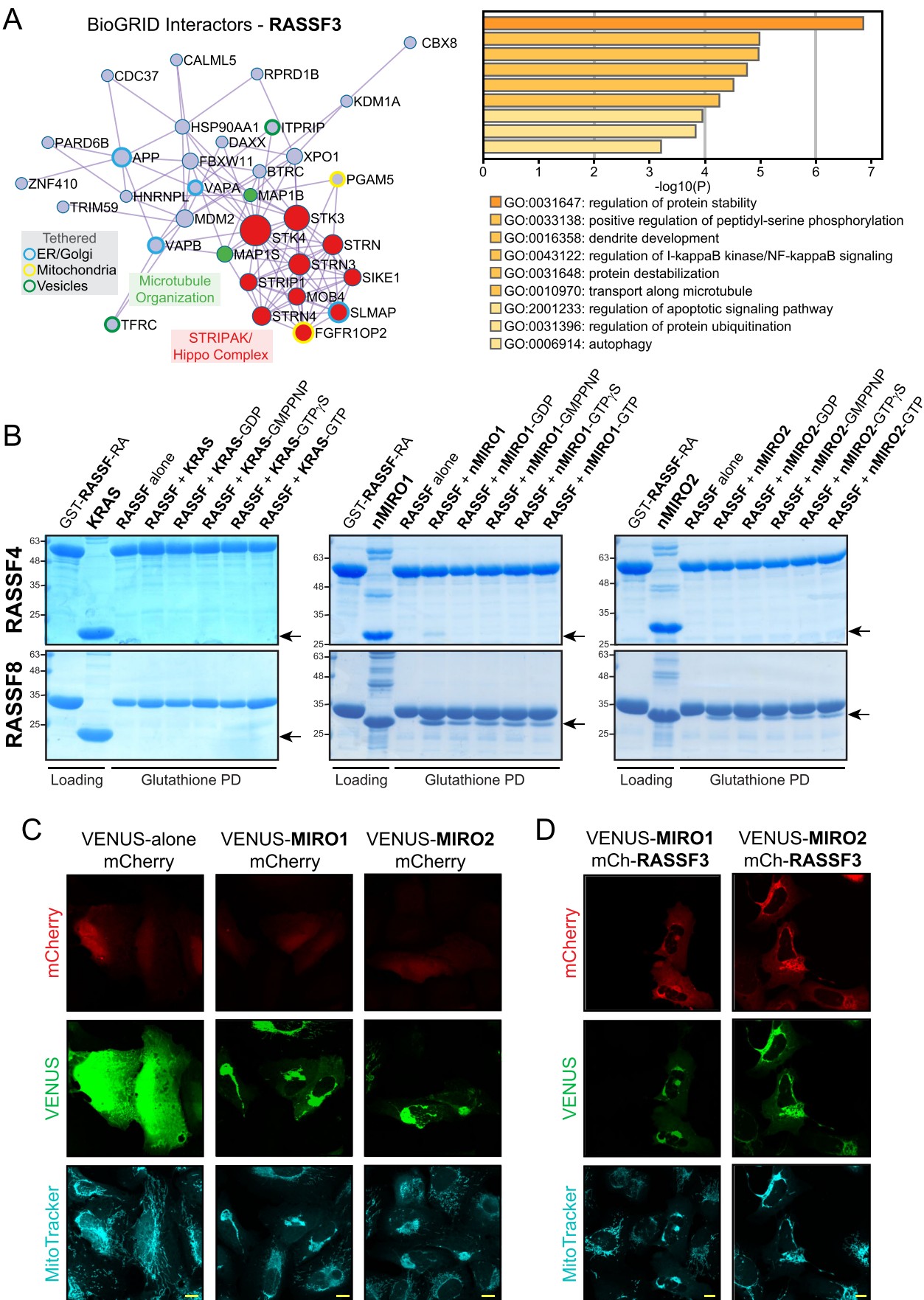

A   BioGRID Interactors - RASSF3

◄  **Figure EV5.   Interaction between RASSF3 and the mitochondrial GTPases MIRO1/2.**

(A) Enrichment analysis of RASSF3 interactors from the BioGRID database. Metascape identified the STIPAK complex and numerous proteins tethered to organelles in the dataset (left). Enriched GO terms are listed at right. Bars are colored by increasing $P$ values as determined by Metascape (using a hypergeometric test and Benjamini–Hochberg $p$ value correction algorithm). (B) In vitro mixing assays using recombinantly purified, GST-tagged RA domains from RASSF4 (top) and RASSF8 (bottom) to precipitate the small GTPase KRAS (left), nGTPase domain of MIRO1 (nMIRO1, middle) or nGTPase domain of MIRO2 (nMIRO2, right) on glutathione beads. Arrow indicates where the precipitated GTPases should appear if bound. GTPases were pre-loaded with the nucleotides indicated at top. (C) Representative confocal microscopy images of collapsed mitochondrial networks in U2OS cells expressing VENUS-MIRO1 or VENUS-MIRO2 with mCherry alone. Scale bars represent 10 μm. (D) Confocal images of collapsed mitochondrial networks in U2OS cells co-expressing VENUS-MIRO1 or VENUS-MIRO2 with mCherry-RASSF8. Scale bars represent 10 μm.

