## [Peer Review File · EMBO Reports]

Complex interplay between RAS GTPases and RASSF effectors regulates subcellular localization of YAP

Swati Singh, Gabriela Bernal Astrain, Ana Maria Hincapie, Marilyn Goudreault and Matthew J. Smith

Corresponding author(s): Matthew Smith (matthew.james.smith@umontreal.ca)

Review Timeline:

Submission Date:	6th Dec 23
Editorial Decision:	1st Feb 24
Revision Received:	10th May 24
Editorial Decision:	6th Jun 24
Revision Received:	20th Jun 24
Accepted:	25th Jun 24

Editor: *Martina Rembold*

Transaction Report:

Dear Dr. Smith

Thank you for the submission of your research manuscript to our journal. I am sorry for the delay in handling your manuscript but we have now received the full set of referee reports that is copied below.

As you will see, the referees acknowledge that the findings are potentially interesting, but they also raise a number of partially overlapping concerns. Since these are stated in the reports, I will not iterate them here but feel that all concerns are pertinent and have to be addressed.

Given the constructive and supportive comments, we would like to invite you to revise your manuscript with the understanding that the referee concerns (as detailed above and in their reports) must be fully addressed and their suggestions taken on board. Please address all referee concerns in a complete point-by-point response. Acceptance of the manuscript will depend on a positive outcome of a second round of review. It is EMBO Reports policy to allow a single round of revision only and acceptance or rejection of the manuscript will therefore depend on the completeness of your responses included in the next, final version of the manuscript.

We realize that it is difficult to revise to a specific deadline. In the interest of protecting the conceptual advance provided by the work, we recommend a revision within 3 months (May 1st). Please discuss the revision progress ahead of this time with the editor if you require more time to complete the revisions.

I am also happy to discuss the revision further via e-mail or a video call, if you wish.

*******IMPORTANT NOTE:**

We perform an initial quality control of all revised manuscripts before re-review. Your manuscript will FAIL this control and the handling will be delayed IN CASE the following APPLIES:

- 1) A data availability section providing access to data deposited in public databases is missing. If you have not deposited any data, please add a sentence to the data availability section that explains that.
- 2) Your manuscript contains statistics and error bars based on $n=2$. Please use scatter blots in these cases. No statistics should be calculated if $n=2$.

When submitting your revised manuscript, please carefully review the instructions that follow below. Failure to include requested items will delay the evaluation of your revision. *****

- 1) a .docx formatted version of the manuscript text (including legends for main figures, EV figures and tables). Please make sure that the changes are highlighted to be clearly visible.
- 2) individual production quality figure files as .eps, .tif, .jpg (one file per figure). Please download our Figure Preparation Guidelines (figure preparation pdf) from our Author Guidelines pages <https://www.embopress.org/page/journal/14693178/authorguide> for more info on how to prepare your figures.
- 3) a .docx formatted letter INCLUDING the reviewers' reports and your detailed point-by-point responses to their comments. As part of the EMBO Press transparent editorial process, the point-by-point response is part of the Review Process File (RPF), which will be published alongside your paper.
- 4) a complete author checklist, which you can download from our author guidelines (<<https://www.embopress.org/page/journal/14693178/authorguide>>). Please insert information in the checklist that is also reflected in the manuscript. The completed author checklist will also be part of the RPF.
- 5) Please note that all corresponding authors are required to supply an ORCID ID for their name upon submission of a revised manuscript (<<https://orcid.org/>>). Please find instructions on how to link your ORCID ID to your account in our manuscript tracking system in our Author guidelines (<<https://www.embopress.org/page/journal/14693178/authorguide#authorshipguidelines>>)
- 6) We replaced Supplementary Information with Expanded View (EV) Figures and Tables that are collapsible/expandable online.

A maximum of 5 EV Figures can be typeset. EV Figures should be cited as 'Figure EV1, Figure EV2' etc... in the text and their respective legends should be included in the main text after the legends of regular figures.

7) Please note that a Data Availability section at the end of Materials and Methods is now mandatory. In case you have no data that requires deposition in a public database, please state so instead of refereeing to the database. See also < <https://www.embopress.org/page/journal/14693178/authorguide#dataavailability>>. Please note that the Data Availability Section is restricted to new primary data that are part of this study.

Additional information on source data and instruction on how to label the files are available <<https://www.embopress.org/page/journal/14693178/authorguide#sourcedata>>.

10) Figure legends and data quantification:

- the name of the statistical test used to generate error bars and P values,
 - the number (n) of independent experiments (please specify technical or biological replicates) underlying each data point,
 - the nature of the bars and error bars (s.d., s.e.m.)
- If the data are obtained from n {less than or equal to} 5, show the individual data points in addition to the SD or SEM.
- If the data are obtained from n {less than or equal to} 2, use scatter blots showing the individual data points.

11) Our journal encourages inclusion of *data citations in the reference list* to directly cite datasets that were re-used and obtained from public databases. Data citations in the article text are distinct from normal bibliographical citations and should directly link to the database records from which the data can be accessed. In the main text, data citations are formatted as follows: "Data ref: Smith et al, 2001" or "Data ref: NCBI Sequence Read Archive PRJNA342805, 2017". In the Reference list, data citations must be labeled with "[DATASET]". A data reference must provide the database name, accession number/identifiers and a resolvable link to the landing page from which the data can be accessed at the end of the reference. Further instructions are available at <<https://www.embopress.org/page/journal/14693178/authorguide#referencesformat>>.

12) All Materials and Methods need to be described in the main text. We would encourage you to use 'Structured Methods', our new Materials and Methods format. According to this format, the Materials and Methods section should include a Reagents and Tools Table (listing key reagents, experimental models, software and relevant equipment and including their sources and relevant identifiers) followed by a Methods and Protocols section in which we encourage the authors to describe their methods using a step-by-step protocol format with bullet points, to facilitate the adoption of the methodologies across labs. More information on how to adhere to this format as well as downloadable templates (.doc or .xls) for the Reagents and Tools Table can be found in our author guidelines: < <https://www.embopress.org/page/journal/14693178/authorguide#manuscriptpreparation>>. An example of a Method paper with Structured Methods can be found here: <<https://www.embopress.org/doi/10.15252/msb.20178071>>.

13) As part of the EMBO publication's Transparent Editorial Process, EMBO Reports publishes online a Review Process File to accompany accepted manuscripts. This File will be published in conjunction with your paper and will include the referee reports, your point-by-point response and all pertinent correspondence relating to the manuscript.

Yours sincerely,

Referee #1:

Singh et al. report a screen for Ras/effector interactions focusing on the RASSF members. Originally thought to act as inducers of apoptosis, RASSF members are now considered to be anti-growth signalling molecules, acting in part through the Hippo pathway. However, as the authors rightly point out, there is little to no information on the regulation and downstream signalling properties of several of the RASSF members. To uncover the landscape of Ras/Rho/Arf GTPase-RASSF interactions, the authors performed a systemic pulldown screen using transient transfections of xx GTPases of the Ras/Rho/Arf families in their GTPase-deficient point mutant versions to GTP-load them against 4 RASSF effectors. In addition to confirming known interactions, the current results reveal novel interactions between RASSF members and Ras and Rho GTPases. Given the importance of RASSF/HIPPO crosstalk, in the first part of the paper the authors collect data on the role of the newly identified GTPase/RASSF interactions with respect to HIPPO pathway activation, monitored as nuclear exclusion of YAP. For the only known Ras effector, RASSF5, the authors document how Ras-dependent recruitment of RASSF5 activates the Hippo pathway. They also focus on RASSF8 and the interaction with MIRO documented here. Singh et al. present data indicating that RASSF8 undergoes liquid-liquid phase separation, coexists with YAP and associates with MIRO in unconventional structures, affecting mitochondrial physiology.

Overall, this is an elegant study based on a good deal of hard and solid cell biology and protein biochemistry work that sheds new light on the properties of RASSF members. At a fundamental level, the work illustrates the complexity of the Ras GTPase effector landscape and documents how the same RASSF effectors interact across GTPase family boundaries with Ras and Rho family members. On a special level, the study describes novel interactions, such as the MIRO/RASSF pairs, which are described in detail. Overall, I think the data justify the conclusion drawn. However, there are a number of issues that I recommend be addressed to make this story completely watertight. In particular, I suggest that the authors should invest more in discussing the issue of binding specificity and affinities with its many implications in the text.

Major Points:

- The authors classify GTPase/RASSF interactions as significant based solely on a Western blot signal. Given the assay used, this is acceptable to a first approximation, but biologically a dubious conclusion. Several of the interactions marked as non-existent are likely to have micromolar affinities and we do not understand the relevance of such 'weak' interactions. Adariani et al (PMID: 33930461) analysed RASSF/Ras interactions in solution and determined affinity constants for many of the interactions described here by Singh et al. Their results make it clear that there are many more bona fide GTPase/Ras interactions than the ones evaluated by the authors, they are just much weaker and probably do not give a signal on Western blot. I suggest that the authors discuss this aspect in more detail and critically, and cite the work of Adariani et al.
- There is a concern about the nucleotide specificity of the reported interactions. By definition, a Ras GTPase effector is characterised by interacting with the GTP-loaded version with higher affinity than with the GDP-loaded form. The only nucleotide-dependent direct comparison data shown in Figure 6D (an unusual assay with a Coomassie readout) does not show a clear, convincing nucleotide dependence, even for the well-established KRas/RASSF5 pair. Looking at the strong Coomassie band in the load, an overload of RasGTPases could be a reason for this. Regardless, the authors need to provide convincing data to show that binding, especially of the newly characterised interaction pairs, is GTP-dependent, at least for some of the selected pairs. This could be done by loading purified GTPases in vitro and performing one of several available in vitro assays,

or alternatively by recapitulating the HEK293 pulldowns comparing directly oncogenic mutants with the wild-type GTPase or even nucleotide free/dominant negative versions analogous to N17Ras. While the latter assays certainly have a number of uncertainties (unknown nucleotide exchange rate for many of the GTPases, etc...), it should at least be tried. Depending on the outcome, if it turns out that binding is not dependent on nucleotide loading status, I do not think it is justified to talk about "canonical effectors", as stated throughout the text.

- Looking at the band shape of some of the pulldown blots, as shown in Figure 6A, the FLAG-IP control blots do not appear to be from the same gel/membrane blot as the a-GFP detection. However, the control for IP bait load only makes sense if it is checked on the same membrane used to read out the interactor (GFP). Can the authors comment on why these are different blots?

Minor points:

- The quality of the immunofluorescence image in Fig.2B is poor, I suggest replacing it for better images. Is the control panel EGFP or VENUS as indicated on the panel?
- The authors present Rap proteins as plasma membrane resident. While there is certainly a pool of PM resident Rap, the Raps are prominently localised to endomembranes. I recommend that these statements be re-evaluated and rephrased.
- Figure EV2B: The redistribution of YAP in cells expressing MRas, Rap2C, Rit1 is not visible to me. Please replace with better figures.
- Figure 4F: typo: sorbitol

Referee #2:

Summary

1. Does this manuscript report a single key finding? NO
2. Is the reported work of significance (YES), or does it describe a confirmatory finding or one that has already been documented using other methods or in other organisms etc (NO)? YES
3. Is it of general interest to the molecular biology community? YES: The manuscript revealed novel interactions of RASSF proteins with GTPases from RAS superfamily, a possible association of RASSF8 to liquid-like phase separated condensates and provided preliminary evidence on the impact of RASSF3-MIRO1/2 interaction on mitochondrial network.
4. Is the single major finding robustly documented using independent lines of experimental evidence (YES), or is it really just a preliminary report requiring significant further data to become convincing, and thus more suited to a longer-format article (NO)? YES

The manuscript submitted by Singh S et al provided new insights into the interaction between RAS superfamily members and RASSF proteins. They were able to monitor interactions between proteins from RAS, RHO and ARF families with RBD and RA domains of BRAF and RASSF proteins, respectively, and they were able to report novel interactions. Among these, they reported an association of RASSF5 with growth-promoting e.g., H/N/KRAS and of RASSF3/4/8 with growth-inhibitory RAS proteins, that are important in the identification of their role in cancer. Moreover, RASSF5 interacts not only with classical H/N/KRAS GTPases, but also with RRAS1/2, MRAS, RAP2B/C and RIT1. These interactions blocked YAP nuclear translocation and thus were considered Hippo pathway activators. Furthermore, they provided data supporting localization of RASSF8 to cytoplasmic condensates together with YAP, and its localization is driven by its coiled-coil domain. Ultimately, another significant message is their discovery of a novel interaction between RASSF3 and MIRO1/2 protein that may be implicated in the mitochondrial network collapse.

Overall, the manuscript provides a step ahead into our understanding of lesser studied RAS family members and identified new interacting partners from RAS superfamily for RASSF proteins, which indicated that the functional cross-talk between RAS and RASSF families may be broader and functional consequences may be deeper than previously known. Also, this significant study provides the starting point for future analyses that will be aimed at clarifying the impact of the abovementioned novel interactions on the whole cell homeostasis and not only to processes associated with a subcellular compartment or subcellular structure. The manuscript is well-written, experimental set ups were in general appropriate to support their conclusions, experimental data is robust and clearly presented, figures are of good quality. However, some additional experiments are either desirable to enhance a message or critical in order to back up the methodology used to draw some important observations and conclusions (detailed in my comments).

Major concerns:

1. Introduction: Authors have mistakenly described in their Introduction that RASSFs contains a domain called "RBD-containing effector proteins". However, this domain is termed RAS-association (RA) domain and it is structurally different than RBDs. This aspect has to be clarified in Introduction and subsequently used as RA throughout the manuscript. Furthermore, it is stated that "... leaving the remaining 9 RASSF effectors with no known GTPase binding partner". This is not correct as Adariani SR et al (PMID: 33930461) demonstrated novel interactions between RRAS1, RIT1, and RALA and RASSF7, RASSF9, and RASSF1, respectively. This has to be changed and addressed accordingly.
2. Authors identified an interaction between RASSF3 and MIRO proteins, which occurs independently on the GDP- and GTP-state, and claimed that "RASSF3 is the first known direct effector of MIRO1/2". Per definition, an effector protein for RAS

superfamily family is considered a protein that binds to effector protein in its active GTP-bound form. Despite contradicting studies in understanding MIRO molecular switch function, data provided clearly indicates binding of RASSF3 to MIRO proteins. However, this binding may occur outside of switch regions and therefore RASSF3 does not discriminate the type of nucleotide bound to MIRO and cannot be termed as effector protein. Therefore, naming as effector in the absence of supporting data has to be changed or tuned down.

3. The identification of a possible location of RASSF8 at LLPS is a major finding and it was demonstrated by FRAP. However, literature related to the LLPS suggests that FRAP in spite of being an important read out method, it should be backed up at least by another assay. Also, authors did not provide images of FRAP assay using as control mCherry alone in Fig 4B, as it is an critical control for LLPS studies.

4. In addition to point 3, they described a localization of RASSF8 at LLPS and at the same time demonstrated interactions of RASSF8 with e.g. RHOBTB1/2 in punctate-like structures. It may be interesting to test whether RHOBTB1/2 associated with RASSF8 in LLPS and thus their interaction may reveal that RASSF8 could function as an adaptor for localization of RHOBTB to LLPS or by tethering RHOBTB1/2 may affect puncta dynamics, thus strengthening their observation with a functional aspect.

Minor concerns:

1. For immunoblots of samples collected from pull-downs performed in cell lysates it is advisable to provide GST-bait controls (e.g., Ponceau).
2. Results, page 6: The following statement "... proteomics are well suited for identifying protein-protein interactions..." is not clear. Proteomics may reveal complexes composition rather than interaction (or an interaction in a protein complex) when samples that are analyzed are the result of e.g., IP, pull-down in cell lysates. Also, it is not clear why small GTPases are challenging to identify due to their small size. This paragraph has to be written in a clearer and precise way.
3. Authors indicated that RASSF5-RA binds KRAS in a nucleotide-dependent manner and did not strongly associate with MIRO1/2. Judging by their qualitative assessment via CBB staining presented in Fig. 6D, when compared to KRAS, especially MIRO1 binding seems stronger in RASSF5+MIRO sample and similar RASSF5+MIRO1-GDP sample. In the absence of a quantitative assessment (e.g., Kd measurements) authors can only claim that MIRO1 is precipitated by RASSF5 and this occurs in a nucleotide-independent manner. Perhaps authors should be more precise in describing these observations.
4. The meaning of mitochondrial collapse is not clear to me, is there any speculation that authors can provide? Is it related to a biological process e.g., mitophagy, is mitochondrial integrity and functionality impaired, may mitochondrial bioenergetics be changed? Is it also possible that a mitochondrial fragmentation occurs in response to MIRO1v4+RASSF3 overexpression? Though not absolutely necessary, additional functional assay may only improve the impact of this observation.
5. In Fig. EV5C, cells overexpressing MIRO1/2 seem to be smaller than control cells, but not as visible in the main Fig. 7 despite some differences are still present. Is MIRO1/2 leading to cytoskeleton rearrangements and mitochondrial collapse is resulting from this process and perhaps the MIRO linker function of mitochondria to microtubules is only a secondary effect?
6. There are some contradictions in the pull-down assays regarding the GST control. In Recombinant protein PDs of GTPases in mammalian lysates authors mention the following: "The intensities from precipitation of EGFP-tagged GTPases by GST-alone were subtracted from intensities observed for GTPase binding with RBDs to control for non-specific binding." At the same time, in Appendix figures legend the following text is found "Interacting bands marked with orange on these representative blots (above) were consistently precipitated by GST alone and should be considered non-specific." These contradict each other and authors have to provide an information about their choice to include or exclude from confirmation of interactions.
7. Authors claim that RASSF3 undergoes peroxisomal relocation upon MIRO1v4 overexpression, but data is complete in the absence of a staining for a peroxisomal marker, although previously published data indicated that v4 splice variant locates at peroxisomes.
8. Abbreviations have to be checked and provided full name where they appear for the first time: e.g., RAS, PI3K, ELMO, TEAD, etc.
9. Missing citations: Page 3, description of H/N/KRAS. Page 3, see above Major point 2. Page 3, description of interactions of RASSF1 with non-canonical GTPases. Page 6, information describing the affinity of RAF kinases and KRAS. Page 7, information describing the affinity of RASSF5 to KRAS.
10. Page 4, second row, "... inhibit THEIR expression"?
11. Page 6: "... absolute requirement to have them in a GTP-loaded state" This statement is not valid unless one aims at studying interaction with effector proteins. As GDP-bound GTPases interact with other proteins, perhaps authors need to clarify and be precise e.g., whether they mean with effector proteins, and in this case they are right.
12. Page 20, Purification of recombinant proteins: missing nGTPase amino acid number to set the domain boundaries; also, for KRAS, to be clarified whether full length or G-domain only was used.
13. MitoTracker has to be checked throughout the text and figures and it has to be spelled according to commercial name. Provider has to be mentioned in Materials and Methods.
14. Page 22, Quantification of puncta or collapsed mitochondrial network. Do triplicates mean technical replicates or independent biological replicates/experiments?
15. Page 30, Fig. 3E legend, missing word „endogenous" for YAP staining.
16. Page 34, Fig. 2B, Venus (inside upper left panel) or EGFP (side marking for upper panels), to be clarified.
17. Page 36, Fig. 4F, Sorbitol or Sorbitol? In figure legend and text used sorbitol, in the figure sorbitol.

Referee #3:

This study by Singh et al. explores interactions between BRAF and RASSF effectors, revealing plasticity in RASSF binding. The RASSF5-RAS GTPase complex activates Hippo signaling, relocating YAP to the cytosol. RASSF8 undergoes liquid-liquid phase separation in YAP-associated condensates, engaging multiple GTPases. Notably, RASSF3 is identified as the first canonical effector of mitochondrial MIRO proteins, impacting mitochondrial distribution. These findings provide insights into the intricate nature of GTPase-effector interactions and cell signaling.

This is a nicely carried out study; its results would interest the larger scientific community working in this area. I only have a couple of minor comments that the authors can consider while revising the manuscript:

Page13

MIRO proteins contain two GTPase domains. How similar are the two GTPase domains? Do you expect the cMIRO1-GTPase to be active towards RASSF3 and 4 like nMIRO1-GTPase? As I understand from the manuscript, the MIRO domains were challenging to work with, but can you try expressing the C-terminal MIRO GTPase domain and check whether it bound RASSF3 and 4 as well, or make an EGFP-Delta-nGTPase MIRO1/2 and check binding that way?

Page6-7

BRAF did not strongly associate any of the 80 other RAS....

should be

BRAF did not strongly associate with any of the 80 other RAS....

Page7

Please provide a reference for the RASSF5-KRAS affinity of 1.7uM.

Page7

"Interestingly, the RBD of the tumor suppressor RASSF5 interacted only with growth-promoting RAS GTPases."

Could you discuss the reason in the discussion? Could it be in a normal cell, RASSF5 counteracts, dampens, or a negative feedback mechanism for RAS-RAF-MEK signaling?

The revised manuscript addresses all referee comments with either new experimental data or further clarification as required. We have added or replaced the following results to the manuscript:

- High resolutions microscopy images for Figures 2B, 2C and 2D.
- Co-precipitation of active/dominant negative mutational variants of VENUS-tagged GTPases with GST-RASSF5-RA in Figure 2A
- Loading for the pull-down assay for VENUS-tagged GTPases using recombinantly purified GST-RASSF5-RA or GST-alone in Figure EV1.C
- High-resolution microscopy images showing endogenous YAP staining in U2OS cells expressing FLAG-RASSF5 alone or with mutationally activated VENUS-MRAS, -RAP2C or -RIT1 in Figure EV2.B
- FRAP assay in cells expressing mCherry-RASSF8 with VENUS-RASD1 or VENUS-RHOBTB1 in Figure EV4.D and EV4.E
- Co-immunoprecipitation of VENUS-tagged MIRO1-cGTPase domain or MIRO2-cGTPase with FLAG-RASSF3, FLAG-RASSF4, FLAG-RASSF5, FLAG-RASSF8 and FLAG-alone in Figure 6D
- Confocal images of U2OS cells co-expressing EGFP-MIRO1v4 and mCherry-RASSF3 stained with a peroxisomal marker (PEX14) in Figure 7D

Specific comments to each reviewer follow.

Referee #1

Singh et al. report a screen for Ras/effector interactions focusing on the RASSF members. Originally thought to act as inducers of apoptosis, RASSF members are now considered to be anti-growth signalling molecules, acting in part through the Hippo pathway. However, as the authors rightly point out, there is little to no information on the regulation and downstream signalling properties of several of the RASSF members. To uncover the landscape of Ras/Rho/Arf GTPase-RASSF interactions, the authors performed a systemic pulldown screen using transient transfections of xx GTPases of the Ras/Rho/Arf families in their GTPase-deficient point mutant versions to GTP-load them against 4 RASSF effectors. In addition to confirming known interactions, the current results reveal novel interactions between RASSF members and Ras and Rho GTPases. Given the importance of RASSF/HIPPO crosstalk, in the first part of the paper the authors collect data on the role of the newly identified GTPase/RASSF interactions with respect to HIPPO pathway activation, monitored as nuclear exclusion of YAP. For the only known Ras effector, RASSF5, the authors document how Ras-dependent recruitment of RASSF5 activates the Hippo pathway. They also focus on RASSF8 and the interaction with MIRO documented here. Singh et al. present data indicating that RASSF8 undergoes liquid-liquid phase separation, coexists with YAP and associates with MIRO in unconventional structures, affecting mitochondrial physiology.

Overall, this is an elegant study based on a good deal of hard and solid cell biology and protein biochemistry work that sheds new light on the properties of RASSF members. At a fundamental level, the work illustrates the complexity of the Ras GTPase effector landscape and documents how the same RASSF effectors interact across GTPase family boundaries with Ras and Rho family members. On a special level, the study describes novel interactions, such as the MIRO/RASSF pairs, which are described in detail. Overall, I think the data justify the conclusion drawn. However, there are a number of issues that I recommend be addressed to make this story completely

watertight. In particular, I suggest that the authors should invest more in discussing the issue of binding specificity and affinities with its many implications in the text.

We thank the reviewer for highlighting the significance of this study and for the thoughtful suggestions. We have attempted to address all the points raised, including by adding more discussion of binding affinity/specificity to the written text.

Major Points:

1) The authors classify GTPase/RASSF interactions as significant based solely on a Western blot signal. Given the assay used, this is acceptable to a first approximation, but biologically a dubious conclusion. Several of the interactions marked as non-existent are likely to have micromolar affinities and we do not understand the relevance of such 'weak' interactions. Adariani et al (PMID: 33930461) analysed RASSF/Ras interactions in solution and determined affinity constants for many of the interactions described here by Singh et al. Their results make it clear that there are many more bona fide GTPase/Ras interactions than the ones evaluated by the authors, they are just much weaker and probably do not give a signal on Western blot. I suggest that the authors discuss this aspect in more detail and critically, and cite the work of Adariani et al.

Adariani et al investigated the binding affinities of MBP-tagged RA domains from 10 RASSF homologues (and CRAF) towards 7 RAS GTPases (HRAS, RRAS1, RIT1, RAP1B, RAP2A, RALA and RHEB) using bacterially purified proteins. In their study, CRAF-RBD exhibited the highest affinity for HRAS and RRAS1, while RASSF5 bound with high affinity to HRAS, RAP1B and RAP2A (1-4 μ M). In addition, the RA domain of RASSF5 bound RRAS, RALA and RHEB with weak affinity ($>45 \mu$ M). Other interactions exhibited low (31- 90 μ M) or very weak (91-510 μ M) affinities in their screen. In our work, we performed pull-down assays in a high-throughput manner to systematically map the interaction of RBD/RAs from BRAF, RASSF3, RASSF4, RASSF5 and RASSF8 with all small GTPases of the RAS, RHO and ARF subfamilies. In contrast to the referenced study, we did not observe an interaction between RASSF5 and RAP1B but did uncover binding between RASSF5 and RIT1. Differences in specificity are likely derived from the fact we used full length proteins expressed in human cells (with all available post-translational modifications) rather than bacterially expressed proteins – which, as we demonstrate below, can be challenging to work with. As we were able to detect a RASSF5-RRAS complex measured at 56 μ M by Adariani *et al*, as well as RASSF4-RIT1 binding (58 μ M), this suggests our approach uncovered moderate-to-low affinity interactions. This is typical of pull-down approaches that are expected to perform well in this range. For very weak interactions ($>100 \mu$ M), we agree with the reviewer's statement that these could be missed by our assay. It is generally true that the biological impact of weak interactions has been poorly studied, and for small GTPases further investigation is required concerning the role of low-affinity complexes such as those identified by Adariani *et al*. How these weak binders manifest in a cellular context and how such interactions might compete with high-affinity GTPase-effector complexes is unknown. We have added this point to our Discussion and included the Adariani *et al* reference in our revised manuscript.

2) There is a concern about the nucleotide specificity of the reported interactions. By definition, a Ras GTPase effector is characterised by interacting with the GTP-loaded version with higher affinity than with the GDP-loaded form. The only nucleotide-dependent direct comparison data shown in Figure 6D (an unusual assay with a Coomassie readout) does not show a clear, convincing nucleotide dependence, even for the well-established KRas/RASSF5 pair. Looking at the strong Coomassie band in the load, an overload of RasGTPases could be a reason for this.

Regardless, the authors need to provide convincing data to show that binding, especially of the newly characterised interaction pairs, is GTP-dependent, at least for some of the selected pairs. This could be done by loading purified GTPases *in vitro* and performing one of several available *in vitro* assays, or alternatively by recapitulating the HEK293 pulldowns comparing directly oncogenic mutants with the wild-type GTPase or even nucleotide free/dominant negative versions analogous to N17Ras. While the latter assays certainly have a number of uncertainties (unknown nucleotide exchange rate for many of the GTPases, etc...), it should at least be tried. Depending on the outcome, if it turns out that binding is not dependent on nucleotide loading status, I do not think it is justified to talk about "canonical effectors", as stated throughout the text.

As pointed out by the reviewer, the defining characteristic of a RAS effector lies in its ability to bind RAS GTPases in a nucleotide-dependent manner. Of course, this definition is built on the archetypal RAS-effector interactions and may not be a defining characteristic of all small GTPase-effector partners. Nevertheless, we certainly agree this is an avenue worth exploring and made significant attempts at studying the direct interactions *in vitro* (as shown in the initial submission for MIRO interactions). Performing this with nearly 50 novel complexes and a highly diverse set of largely unstudied GTPase proteins quickly became a huge challenge, particularly because unlike H/K/NRAS the vast majority of purified small GTPases proved highly unstable in solution. We have been unable to purify many of the GTPases of interest to high concentration and homogeneity (for selective nucleotide loading and/or NMR-based analyses) including RASL11A, RASL11B, RIT1, RASL12, RASD1, RASD2 and MIRO (**Revision Figure 1/2**). Moreover, we attempted to purify the MIRO-nGTPase domain from different species including *Xenopus* (frog), chicken and mouse but all proved insoluble (**Revision Figure 2**). Our group has over 15 years experience producing a wide range of small GTPase proteins and mutational variants for biophysical and structural analyses (Thillaivillalan *et al*, 2020; Chang *et al*, 2020; Killoran & Smith, 2019; Smith *et al*, 2017; Smith & Ikura, 2014; Smith *et al*, 2013; Findlay *et al*, 2013) and we have tried multiple different tagging and purification strategies with these, but most of these enigmatic G-proteins proved extremely difficult to work with. We have recently published data on ARF GTPases (Quirion *et al*, 2024) that suggests 1) weak nucleotide affinity may underlie poor solubility, and 2) not all small GTPases behave as canonical 'switch' proteins and it can be misleading to assume they function as such. For the current study, it was only following numerous troubleshooting attempts over the course of 2-3 years that we successfully derived conditions to purify the human MIRO-nGTPase domain (**Revision Figure 3**) for use in the binding studies detailed in our manuscript. Still, the nGTPase domains of both MIRO1 and MIRO2 exhibit significant instability in solution and mixing either of these with the RASSF3-RA domain exacerbated this for all biophysical assays, including NMR, and we were therefore unable to perform isothermal titration calorimetry (ITC) to determine binding affinities. Instead, we chose to perform *in vitro* mixing assays utilizing recombinantly purified proteins and retain the GST tag to show direct binding of RASSF RA domains with KRAS or MIRO. Notably, RASSF5 did bind KRAS in a nucleotide-dependent manner as evidenced by enhanced binding observed with KRAS-GTP γ S or KRAS-GTP (Figure 6E), though we did pick up a low affinity complex with KRAS-GDP. This is not unusual, nearly every immunoprecipitation or pull-down assay with small GTPases and RBDs in the literature reveals not a complete loss of binding to the GDP-bound protein, but rather significantly reduced binding (the high avidity effect from the bead-immobilized RBD will capture low affinity interactions: RAF binds RAS-GDP with a K_d of 46 μ M (Kiel *et al*, 2009)).

For all proteins: Cloned in pGEX-4T-1, transformed in BL21-DE3 and grown in M9 media, induction with 250uM of IPTG followed with overnight incubation at 16°C

Revision Figure 1. Affinity purification of GST-tagged full-length (FL) RASL11A, RASL11B, RIT1 and RASL12 RAS-subfamily GTPases. These GTPases were transformed into BL21-DE3 *E. coli* cells and grown here in M9 minimal media. Protein expression was induced using 250 μ M IPTG and cells were grown at 16 °C for 16 hours after induction. Proteins were highly unstable following cleavage of the GST-tag and immediately precipitated, as indicated by the lack of detectable protein in the Cleaved protein lane (boxed).

Revision Figure 2. Expression and solubility check for MIRO orthologs and the RASD1/2 small G-proteins. GTPases were cloned in pDEST17 bacterial expression system and transformed into BL21-A1 *E. coli* cells. Expression was induced using the indicated percentage of L-Arabinose and grown at 16 °C for 16 hours after induction. Cells were lysed and centrifuged at 15000 rpm for 15 mins. Pellet indicates total cell lysate, the Soluble lane is supernatant and the Insoluble lane shows protein is in the pellet following centrifugation. Arrows show all GTPases are predominantly insoluble under these expression and purification conditions.

Revision Figure 3. Recombinant protein purification using affinity purification (Nickel-NTA) and gel filtration of His-tagged Human MIRO1-nGTPase and MIRO2-nGTPase.

Complete characterization of our identified complexes *in vitro* will therefore be a massive undertaking and is beyond the scope of the current work. However, to provide additional support for our observations, and at the reviewer’s suggestion, we performed pull-down assays with GST-RASSF5 RA domain and mutational variants of KRAS (wild-type, constitutively active (G12V and Q61L) and dominant negative (S17N)) together with corresponding mutations in RRAS1 and RIT1 (new Figure 2A). The RASSF5 RA domain demonstrated diminished binding to wild-type KRAS and RRAS1, though bound strongly to wild-type RIT1. No binding to the dominant negative mutants was observed with any of the GTPases (KRAS-S17N, RRAS1-S27N and RIT1-S35N). Conversely, RASSF5 exhibited strong binding to the presumed GTP-loaded mutants G12V and Q61L of KRAS, as well as analogous RRAS1 (G30V and Q87L) and RIT1 (G30V and Q79L) mutants. This demonstrates that RASSF5 binds distinct RAS subfamily GTPases in a nucleotide-dependent manner. We must state that this approach comes with many caveats and is not nearly as robust as performing *in vitro* experiments with isolated proteins of defined nucleotide-loaded states. It relies on an assumption that all small GTPases function in completely the same way and are regulated in cells in a manner analogous to H/K/NRAS, which is clearly not true. Recent studies suggest that some GTPases lack hydrolysis activity and are pseudoGTPases (eg RHOBTB1 and RHOBTB2, which emerged as candidate partners for RASSF8 in our study (Stiegler & Boggon, 2020)). The crystal structure of MIRO1-nGTPase (PDBid 6D71) is in a GTP-loaded state using protein directly purified from *E. coli*, suggesting that MIRO cannot hydrolyze GTP (Smith *et al*, 2020). Our recent ARF work demonstrates that several members of this subfamily do not load nucleotides in a canonical fashion (Quirion *et al*, 2024), [REDACTED: Author's response with unpublished data.]. There is limited biochemical data available on nucleotide-cycling for the majority of RAS superfamily GTPases. As such, we have addressed the reviewer’s point in our Discussion to include concerns that the identified effector candidates may not function as ‘canonical’ effectors, but do not believe attempting full panels of pull downs with artificially designed mutants of unknown biochemical function would add significantly to the data already present.

3) Looking at the band shape of some of the pulldown blots, as shown in Figure 6A, the FLAG-IP control blots do not appear to be from the same gel/membrane blot as the a-GFP detection. However, the control for IP bait load only makes sense if it is checked on the same membrane used to read out the interactor (GFP). Can the authors comment on why these are different blots?

The objectives of these IP experiments were to test RASSF3 binding with different GTPases identified as binding partners in the pull-down screen. We agree that detecting the bait load on the same membrane used to readout the prey is optimal, yet in some cases this approach is not practical: eg, if the bait and prey are identical in molecular weight (MW). This was the case here, as FLAG-RASSF3 (30 kDa) and FLAG-RASSF4 (48 kDa) run at the same MW as several of the candidate GTPases. We therefore opted not to strip the blots and instead loaded equal amounts of the identical IP onto two individual SDS-PAGE gels to perform Western Blotting.

Minor points:

4) The quality of the immunofluorescence image in Fig. 2B is poor, I suggest replacing it for better images. Is the control panel EGFP or VENUS as indicated on the panel?

We have replaced the immunofluorescence panels in Figure 2B/C/D with higher quality images. We have corrected labelling for VENUS.

5) The authors present Rap proteins as plasma membrane resident. While there is certainly a pool of PM resident Rap, the Raps are prominently localised to endomembranes. I recommend that these statements be re-evaluated and rephrased.

Indeed, RAP2 GTPases can be localized both at the plasma membrane and endosomes (Ohba *et al*, 2000; Uechi *et al*, 2009; Paganini *et al*, 2006). We focused here on plasma membrane localization, as RASSF5 notably was recruited to the PM when co-expressed with RAP2B or RAP2C. Nonetheless, we have rephrased the sentence to include potential endosomal localization of RAP GTPases.

6) Figure EV2B: The redistribution of YAP in cells expressing MRas, Rap2C, Rit1 is not visible to me. Please replace with better figures.

We have incorporated new immunofluorescence images in Figure EV2B showing endogenous YAP in cells co-expressing RASSF5 with MRAS, RAP2C and RIT1.

7) Figure 4F: typo: sorbitiol

We have corrected this typo in Figure 4F.

Referee #2

The manuscript submitted by Singh S et al provided new insights into the interaction between RAS superfamily members and RASSF proteins. They were able to monitor interactions between proteins from RAS, RHO and ARF families with RBD and RA domains of BRAF and RASSF proteins, respectively, and they were able to report novel interactions. Among these, they reported an association of RASSF5 with growth-promoting e.g., H/N/KRAS and of RASSF3/4/8 with growth-inhibitory RAS proteins, that are important in the identification of their role in cancer. Moreover, RASSF5 interacts not only with classical H/N/KRAS GTPases, but also with RRAS1/2, MRAS, RAP2B/C and RIT1. These interactions blocked YAP nuclear translocation and thus were considered Hippo pathway activators. Furthermore, they provided data supporting localization of

RASSF8 to cytoplasmic condensates together with YAP, and its localization is driven by its coiled-coil domain. Ultimately, another significant message is their discovery of a novel interaction between RASSF3 and MIRO1/2 protein that may be implicated in the mitochondrial network collapse.

Overall, the manuscript provides a step ahead into our understanding of lesser studied RAS family members and identified new interacting partners from RAS superfamily for RASSF proteins, which indicated that the functional cross-talk between RAS and RASSF families may be broader and functional consequences may be deeper than previously known. Also, this significant study provides the starting point for future analyses that will be aimed at clarifying the impact of the abovementioned novel interactions on the whole cell homeostasis and not only to processes associated with a subcellular compartment or subcellular structure. The manuscript is well-written, experimental set ups were in general appropriate to support their conclusions, experimental data is robust and clearly presented, figures are of good quality. However, some additional experiments are either desirable to enhance a message or critical in order to back up the methodology used to draw some important observations and conclusions (detailed in my comments).

We thank the reviewer for their thoughtful comments. We have answered all of them below and incorporated many of the suggested changes to the manuscript.

Major concerns:

1) Introduction: Authors have mistakenly described in their Introduction that RASSFs contains a domain called "RBD-containing effector proteins". However, this domain is termed RAS-association (RA) domain and it is structurally different than RBDs. This aspect has to be clarified in Introduction and subsequently used as RA throughout the manuscript. Furthermore, it is stated that "... leaving the remaining 9 RASSF effectors with no known GTPase binding partner". This is not correct as Adariani SR et al (PMID: 33930461) demonstrated novel interactions between RRAS1, RIT1, and RALA and RASSF7, RASSF9, and RASSF1, respectively. This has to be changed and addressed accordingly.

We have previously solved GTPase-effector RBD crystal structures (Smith *et al*, 2017; Chang *et al*, 2020) and are not aware of any "structural differences" between RA and RBD domains. Nevertheless, we have changed all the text and Figure labels to reflect this historical naming distinction at the reviewer's request. We have also added further discussion of the Adariani et al paper (see response to Reviewer 1, point 1).

2) Authors identified an interaction between RASSF3 and MIRO proteins, which occurs independently on the GDP- and GTP- state, and claimed that "RASSF3 is the first known direct effector of MIRO1/2". Per definition, an effector protein for RAS superfamily family is considered a protein that binds to effector protein in its active GTP-bound form. Despite contradicting studies in understanding MIRO molecular switch function, data provided clearly indicates binding of RASSF3 to MIRO proteins. However, this binding may occur outside of switch regions and therefore RASSF3 does not discriminate the type of nucleotide bound to MIRO and cannot be termed as effector protein. Therefore, naming as effector in the absence of supporting data has to be changed or tuned down.

The current paradigm that effectors bind to small GTPases in a GTP-dependent manner is an archetype derived from intense study of a small subset of RAS proteins (eg H/K/NRAS and

RHO/RAC/CDC42). This may not prove relevant to all small G-proteins as an increasing number do not function as canonical ‘switch-like’ proteins and are considered pseudoGTPases. This includes many of the tumour-suppressor class RAS proteins revealed as hits in this work (we have reviewed the biochemical function of these proteins (Bernal Astrain *et al*, 2022)) and may include MIRO, which has been demonstrated to hydrolyse GTP (Peters *et al*, 2018) yet crystallized in a GTP-bound state suggesting it lacks GTPase activity (Smith *et al*, 2020). It is therefore possible that MIRO may interact with its effector proteins in a nucleotide-independent manner, or that some factor(s) required for nucleotide switching (eg PTMs) is lacking from bacterially-expressed protein. As discussed in our response to Reviewer 1 (point 2) we made significant attempts to purify and characterize many of the understudied GTPases of interest identified by the screen, but they have proven very difficult to isolate *in vitro*. Nonetheless, the reviewer is correct that binding between RASSF3 and MIRO may occur outside of switch regions or be independent of nucleotide loading. We have rephrased the sentence to show RASSF3 as a “potential” effector of MIRO1-nGTPase and now discuss the canonical nature of these interactions in the Discussion.

3) *The identification of a possible location of RASSF8 at LLPS is a major finding and it was demonstrated by FRAP. However, literature related to the LLPS suggests that FRAP in spite of being an important read out method, it should be backed up at least by another assay. Also, authors did not provide images of FRAP assay using as control mCherry alone in Fig 4B, as it is an critical control for LLPS studies.*

Figure 4B shows FRAP of RASSF8 condensates to demonstrate LLPS. FRAP is the current gold-standard assay to demonstrate LLPS in cells, and combined with validation that individual condensates can fuse (also shown in Figure 4B) this constitutes the two leading experimental approaches. The identical methodologies were used in recent studies establishing that proteins of the Hippo pathway undergo phase separation (Wang *et al*, 2022; Bonello *et al*, 2023). We contacted a leading researcher in the LLPS field to determine if a substitute approach could/should be used to demonstrate that these are condensates in cells. We were informed that alternative methods are currently limited to:

1. *Fluorescence Correlation Spectroscopy (FCS)*: this measures fluctuation of protein molecules in the condensate to calculate a partition coefficient, similar to FRAP but potentially providing a diffusion coefficient with appropriate microscope setup; unfortunately, we do not have access to the appropriate microscope to perform this work and it is more complementary to FRAP than a distinctive demonstration of LLPS

2. *Single Particle Tracking (SPT)*: to measure a diffusion coefficient and show that molecules are moving relatively slower in the condensate than the surrounding environment; this again requires a high level of expertise and appropriate microscopy infrastructure for which we lack access

These two alternative approaches may provide detailed kinetic data on particle motion in cells, and whether the movement of proteins is slower in condensates than in outside regions, but aren’t novel demonstrations of LLPS. We would be happy to perform other supplementary experiments if the reviewer has suggestions. An additional supporting approach entails *in vitro* purification of fluorescently tagged proteins and demonstration of LLPS induced by changing salt/temperature conditions. We attempted to purify full length RASSF8, but it was highly insoluble. These are long and demanding projects individually and we believe the demonstration of LLPS in the manuscript, particularly in context of the recent work with Hippo proteins, is highly conclusive.

For the mCherry alone control – this is not a typical control as the fluorescent protein alone is in a

homogenous phase leading to spontaneous and rapid recovery after photobleaching. Indeed, we performed FRAP on mCherry and were not able to observe significant photobleaching of assigned regions as the protein moves very rapidly through the cytosol with no restrictions.

4) In addition to point 3, they described a localization of RASSF8 at LLPS and at the same time demonstrated interactions of RASSF8 with e.g. RHOTB1/2 in punctate-like structures. It may be interesting to test whether RHOTB1/2 associated with RASSF8 in LLPS and thus their interaction may reveal that RASSF8 could function as an adaptor for localization of RHOTB to LLPS or by tethering RHOTB1/2 may affect puncta dynamics, thus strengthening their observation with a functional aspect.

We thank the reviewer for their suggestion. RASSF8 co-localises with its GTPase partners such as RASD1 or RHOTB1 in irregularly shaped perinuclear structures (sometimes observed throughout the cytosol). We performed FRAP assays for mCherry-RASSF8 co-expressed with GFP-RASD1 or GFP-RHOTB1 and found a much-reduced fluorescence recovery after photobleaching, characteristic of a gel-like state. This indicates that interaction with GTPase partners does indeed change the dynamics of RASSF8 condensates and drives a liquid-to-solid phase transition. A similar transition has been observed for many protein condensates associated with neurological disorders such as α -synuclein, FUS and TDP-43 (Patel *et al*, 2015; Carey & Guo, 2022; Ray *et al*, 2020) and this gel-like state is an emerging but still enigmatic property in cells. We have added the new Figure EV4D/E for FRAP with RASSF8 co-expressed with RASD1 or RHOTB1 and discuss the implications in the text.

Minor concerns:

1) For immunoblots of samples collected from pull-downs performed in cell lysates it is advisable to provide GST-bait controls (e.g., Ponceau).

At the reviewer's suggestion we have added GST-bait loading in Figure 2A and EV1C. For the high-throughput pull-down screens performed to map RAF/RASSF interactions with 83 GTPases from the RAS, RHO and ARF subfamilies, we ensured GST-bait was present for each pull-down on Western blotting (using ponceau stain) but did not capture images.

2) Results, page 6: The following statement "... proteomics are well suited for identifying protein-protein interactions..." is not clear. Proteomics may reveal complexes composition rather than interaction (or an interaction in a protein complex) when samples that are analyzed are the result of e.g., IP, pull-down in cell lysates. Also, it is not clear why small GTPases are challenging to identify due to their small size. This paragraph has to be written in a clearer and precise way.

We have rephrased the sentence.

3) Authors indicated that RASSF5-RA binds KRAS in a nucleotide-dependent manner and did not strongly associate with MIRO1/2. Judging by their qualitative assessment via CBB staining presented in Fig. 6D, when compared to KRAS, especially MIRO1 binding seems stronger in RASSF5+MIRO sample and similar RASSF5+MIRO1-GDP sample. In the absence of a quantitative assessment (e.g., Kd measurements) authors can only claim that MIRO1 is precipitated by RASSF5 and this occurs in a nucleotide-independent manner. Perhaps authors should be more precise in describing these observations.

We appreciate the reviewer's feedback. We did not observe any binding of RASSF5 to MIRO2 and only minimal binding of RASSF5 to MIRO1 (in the absence of nucleotide manipulation), as shown above (highlighted by red box). Unfortunately, the MIRO proteins were difficult to study *in vitro* which precluded measurements of affinity (see response to Reviewer 1, point 2). We agree that this *in vitro* mixing assay does not offer a quantitative evaluation of binding affinity and have attempted to describe these more qualitative results in the text to the best of our ability.

4) *The meaning of mitochondrial collapse is not clear to me, is there any speculation that authors can provide? Is it related to a biological process e.g., mitophagy, is mitochondrial integrity and functionality impaired, may mitochondrial bioenergetics be changed? Is it also possible that a mitochondrial fragmentation occurs in response to MIRO1v4+RASSF3 overexpression? Though not absolutely necessary, additional functional assay may only improve the impact of this observation.*

The significance of such a collapsed mitochondrial network, though frequently observed, is still unknown. Similar perinuclear mitochondrial clustering was previously observed upon MIRO or Mitofusin overexpression (Huo *et al*, 2022; Sloat & Hoppins, 2023). Mitofusins (Mfn1 and Mfn2) are outer mitochondrial membrane proteins which interact with MIRO GTPases. Mitochondrial clustering induced by Mfn2 overexpression causes mitochondrial dysfunction and cell death (Huang *et al*, 2007). Charcot-Marie-Tooth syndrome Type 2A (CMT2A)-associated Mfn variants (Mfn1 S329P and Mfn2 S350P) cause mitochondrial perinuclear clusters (Sloat & Hoppins, 2023). Furthermore, disease-driving PARKIN mutants elicit a similar phenotype, i.e. mitochondrial aggregation or clustering around the nucleus (Lee *et al*, 2010). These PARKIN mutants are defective in mitophagy. More research is needed to understand the consequence of mitochondrial clustering on their function and cellular homeostasis. We have not performed additional functional assays to elucidate a potential biological role for the mitochondrial collapse observed for MIRO + RASSF3, as this would entail a wide variety of potential functional assays to explore the nature of the defect. We speculate that RASSF3 interaction with MIRO likely disrupts MIRO connections to microtubules resulting in mitochondrial collapse and have now added all the above points to the Discussion.

5) *In Fig. EV5C, cells overexpressing MIRO1/2 seem to be smaller than control cells, but not as visible in the main Fig. 7 despite some differences are still present. Is MIRO1/2 leading to cytoskeleton rearrangements and mitochondrial collapse is resulting from this process and perhaps the MIRO linker function of mitochondria to microtubules is only a secondary effect?*

We thank the reviewer for this observation. We can speculate that the collapsed mitochondrial network observed in our study is a result of disrupting MIRO connection to microtubules. We have not quantified the size/volume of cells expressing MIRO1/2 or MIRO GTPases along with RASSF3. We would surmise that mitochondrial collapse would result in diminished cell size, rather than small cell size resulting in mitochondrial collapse.

6) *There are some contradictions in the pull-down assays regarding the GST control. In Recombinant protein PDs of GTPases in mammalian lysates authors mention the following: "The intensities from precipitation of EGFP-tagged GTPases by GST-alone were subtracted from intensities observed for GTPase binding with RBDs to control for non-specific binding." At the same time, in Appendix figures legend the following text is found "Interacting bands marked with orange on these representative blots (above) were consistently precipitated by GST alone and should be considered non-specific." These contradict each other and authors have to provide an information about their choice to include or exclude from confirmation of interactions.*

These statements do not contradict and are asserting the same point. Both were meant to make clear that GTPase interactions observed for GST-alone were considered non-specific (eg those “marked with orange on these representative blots” were observed with GST-alone and should not be considered RBD-specific). For any interactions apparent on the GST-alone control, the intensities of these bands were subtracted from the GST-RBD pull downs to allow construction of the heat maps.

7) *Authors claim that RASSF3 undergoes peroxisomal relocation upon MIRO1v4 overexpression, but data is complete in the absence of a staining for a peroxisomal marker, although previously published data indicated that v4 splice variant locates at peroxisomes.*

We appreciate the suggestion provided by the reviewer and have included new data with staining for a peroxisomal marker in cells co-expressing RASSF3 and MIRO1v4 (Figure 7D). Co-expression of RASSF3 and MIRO1v4 results in clustering not just of mitochondria in the perinuclear region but also peroxisomes. It was not possible to perform staining of peroxisomes, mitochondria, RASSF3 and MIRO simultaneously, but we can now speculate that RASSF3 and MIRO1v4 co-expression results in co-collapse of peroxisomes and mitochondria to a perinuclear region. This is supported by the observation that RASSF3 redistributes MIRO1v4 to mitochondrial clusters in Figure 7E. Interestingly, recent studies are highlighting the existence of contact sites between mitochondria and peroxisomes, and overexpression of mitofusins (Mfn1 and Mfn2) can facilitate contact and co-clustering of these organelles (Huo *et al*, 2022; Alsayyah *et al*, 2024; Shai *et al*, 2018).

8) *Abbreviations have to be checked and provided full name where they appear for the first time: e.g., RAS, PI3K, ELMO, TEAD, etc.*

This suggested change has been incorporated.

9) *Missing citations: Page 3, description of H/N/KRAS. Page 3, see above Major point 2. Page 3, description of interactions of RASSF1 with non-canonical GTPases. Page 6, information describing the affinity of RAF kinases and KRAS. Page 7, information describing the affinity of RASSF5 to KRAS.*

We have now included these citations.

10) *Page 4, second row, "... inhibit THEIR expression"?*

This typo has been rectified.

11) Page 6: "... absolute requirement to have them in a GTP-loaded state" This statement is not valid unless one aims at studying interaction with effector proteins. As GDP-bound GTPases interact with other proteins, perhaps authors need to clarify and be precise e.g., whether they mean with effector proteins, and in this case they are right.

On page 6 we mention that "To better understand the specificity of RBD domains and identify binding partners for diverse RASSF proteins, we attempted to map RASSF interactions with all RAS, RHO and ARF subfamily GTPases. Discovery-based approaches such as proteomics are well suited for identifying protein-protein interactions, however, small GTPases are challenging to identify due to their small size and the absolute requirement to have them in a GTP-loaded state." This is therefore in context of identifying GTPase partners for effector RBD/RAs. We gave the limitations in identifying GTPase partners for effectors using discovery-based approach and the reason to map RASSF interactions with GTPase using pull-down assays. We have rephrased the sentence for clarification.

12) Page 20, Purification of recombinant proteins: missing nGTPase amino acid number to set the domain boundaries; also, for KRAS, to be clarified whether full length or G-domain only was used.

Thank you for noticing this. We have incorporated more information in the methods and materials.

13) MitoTracker has to be checked throughout the text and figures and it has to be spelled according to commercial name. Provider has to be mentioned in Materials and Methods.

We have incorporated this change.

14) Page 22, Quantification of puncta or collapsed mitochondrial network. Do triplicates mean technical replicates or independent biological replicates/experiments?

Quantification was done in three independent biological replicates and more than 75 cells were counted for each replicate.

15) Page 30, Fig. 3E legend, missing word „endogenous" for YAP staining.

We have included "endogenous" for YAP immunostaining in Figure 3E legend.

16) Page 34, Fig. 2B, Venus (inside upper left panel) or EGFP (side marking for upper panels), to be clarified.

This is clarified.

17) Page 36, Fig. 4F, Sorbitol or Sorbitol? In figure legend and text used sorbitol, in the figure sorbitol.

This typo is now rectified.

Referee #3

This study by Singh et al. explores interactions between BRAF and RASSF effectors, revealing plasticity in RASSF binding. The RASSF5-RAS GTPase complex activates Hippo signaling, relocating YAP to the cytosol. RASSF8 undergoes liquid-liquid phase separation in YAP-associated condensates, engaging multiple GTPases. Notably, RASSF3 is identified as the first canonical effector of mitochondrial MIRO proteins, impacting mitochondrial distribution. These

findings provide insights into the intricate nature of GTPase-effector interactions and cell signaling.

This is a nicely carried out study; its results would interest the larger scientific community working in this area. I only have a couple of minor comments that the authors can consider while revising the manuscript:

We thank the reviewer for appreciating the significance of our study and for the comments and feedback.

Page13: MIRO proteins contain two GTPase domains. How similar are the two GTPase domains? Do you expect the cMIRO1-GTPase to be active towards RASSF3 and 4 like nMIRO1-GTPase? As I understand from the manuscript, the MIRO domains were challenging to work with, but can you try expressing the C-terminal MIRO GTPase domain and check whether it bound RASSF3 and 4 as well, or make an EGFP-Delta-nGTPase MIRO1/2 and check binding that way?

We have now performed binding assays with the cGTPase domains of these MIRO proteins (not included in our original screen) and found that none of the four RASSF homologues could complex with the MIRO1-cGTPase (see the new Figure 6D). RASSF3, however, can associate with the MIRO2-cGTPase domain and both RASSF4 and RASSF5 also showed binding. This suggests that RASSF3 has a capacity to interact with only the nGTPase domain of MIRO1 and both GTPase domains of MIRO2. We have added this point to the text.

Page6-7: 'BRAf did not strongly associate any of the 80 other RAS....' should be 'BRAf did not strongly associate with any of the 80 other RAS....'

This typo is corrected.

Page7: Please provide a reference for the RASSF5-KRAS affinity of 1.7uM.

The reference for the RASSF5-KRAS affinity has been added.

Page7: "Interestingly, the RBD of the tumor suppressor RASSF5 interacted only with growth-promoting RAS GTPases." Could you discuss the reason in the discussion? Could it be in a normal cell, RASSF5 counteracts, dampens, or a negative feedback mechanism for RAS-RAF-MEK signaling?

The reviewer has brought up an important point for discussion. Yes, we do postulate that RASSF5 would serve as a negative regulator of proliferation in normal cells (eg during development) by enabling crosstalk between the Hippo and RAS pathways. This does not necessitate that RASSF5 expression completely outcompetes or inhibits MAPK activation downstream of RAS. Rather, as work in many human cancer cell lines has demonstrated, RAS-induced proliferation can be contingent on nuclear YAP for reasons that aren't completely understood (Hong *et al*, 2014; Shao *et al*, 2014; Zhang *et al*, 2014; Garcia-Rendueles *et al*, 2015; Slemmons *et al*, 2015; Lin & Bivona, 2016; Justine Pascual, Jelle Jacobs, Leticia Sansores-Garcia, Malini Natarajan, Julia Zeitlinger, Stein Aerts, Georg Halder, 2017; Coggins *et al*, 2019; Tu *et al*, 2019). We expect if RASSF5 is expressed and available for binding activated RAS GTPases that promote cellular proliferation, this will activate the Hippo pathway and sequester YAP in the cytoplasm to constrain growth. The regulation of RASSF5 expression (transcriptionally and post-translationally) could enable this function as a negative regulator, and is expectedly context dependent (cell density, tension, size, adhesion, etc). This is the likely reason that RASSF expression in cultured cell lines is typically low (see Figure 3A). Here, we show that RASSF5 can perform this function downstream of

numerous growth-promoting GTPases. There is one founding member of the RASSF family present in single cell choanoflagellates (eg *Monosiga brevicollis*). This lone RASSF ortholog shares domain organization (C1-RA-SARAH) with RASSF1 (does not bind H/K/NRAS (Thillaivillalan *et al*, 2020)) and RASSF5 (binds H/K/NRAS and similar growth-drivers). Unfortunately, there are no biochemical or functional data on any early orthologs to help delineate which GTPases these proteins bind to or whether they modulate Hippo activity. Pre-metazoans do have extensive pTyr-regulated signalling networks including RTKs (Manning *et al*, 2008), as well as conserved Hippo components (Sebé-Pedrós *et al*, 2012). Taken together, our current submission and our previous work (Thillaivillalan *et al*, 2020) reveal that multiple RASSF paralogs have evolved with increasing animal complexity, each demonstrating a unique binding specificity for an array of RAS superfamily proteins. This provides cells a complex array of negative regulators of cell proliferation that act depending on the activation state of diverse small G-proteins. Substantiating this hypothesis will require significant work *in vitro* and in both simple and complex model organisms. We have now added several of the above points in the Discussion.

References

- Alsayyah C, Singh MK, Morcillo-Parra MA, Cavellini L, Shai N, Schmitt C, Schuldiner M, Zalckvar E, Mallet A, Belgareh-Touzé N, *et al* (2024) Mitofusin-mediated contacts between mitochondria and peroxisomes regulate mitochondrial fusion. *PLOS Biology* 22: e3002602
- Bernal Astrain G, Nikolova M & Smith MJ (2022) Functional diversity in the RAS subfamily of small GTPases. *Biochemical Society Transactions* 50: 921–933
- Bonello TT, Cai D, Fletcher GC, Wiengartner K, Pengilly V, Lange KS, Liu Z, Lippincott-Schwartz J, Kavran JM & Thompson BJ (2023) Phase separation of Hippo signalling complexes. *The EMBO Journal* 42: e112863
- Carey JL & Guo L (2022) Liquid-Liquid Phase Separation of TDP-43 and FUS in Physiology and Pathology of Neurodegenerative Diseases. *Front Mol Biosci* 9: 826719
- Chang L, Yang J, Jo CH, Boland A, Zhang Z, McLaughlin SH, Abu-Thuraia A, Killoran RC, Smith MJ, Côté JF, *et al* (2020) Structure of the DOCK2–ELMO1 complex provides insights into regulation of the auto-inhibited state. *Nature Communications* 11: 1–17
- Coggins GE, Farrel A, Rathi KS, Hayes CM, Scolaro L, Rokita JL & Maris JM (2019) YAP1 mediates resistance to MEK1/2 inhibition in neuroblastomas with hyperactivated Ras signaling. *Cancer Research* 79: 6204–6214
- Findlay GM, Smith MJ, Lanner F, Hsiung MS, Gish GD, Petsalaki E, Cockburn K, Kaneko T, Huang H, Bagshaw RD, *et al* (2013) Interaction Domains of Sos1/Grb2 Are Finely Tuned for Cooperative Control of Embryonic Stem Cell Fate. *Cell* 152: 1008–1020
- Garcia-Rendueles MER, Ricarte-Filho JC, Untch BR, Landa I, Knauf JA, Voza F, Smith VE, Ganly I, Taylor BS, Persaud Y, *et al* (2015) NF2 loss promotes oncogenic RAS-induced thyroid

- cancers via YAP-dependent transactivation of RAS proteins and sensitizes them to MEK inhibition. *Cancer Discovery* 5: 1178–1193
- Hong X, Nguyen HT, Chen Q, Zhang R, Hagman Z, Voorhoeve PM & Cohen SM (2014) Opposing activities of the Ras and Hippo pathways converge on regulation of YAP protein turnover. *EMBO Journal* 33: 2447–2457
- Huang P, Yu T & Yoon Y (2007) Mitochondrial clustering induced by overexpression of the mitochondrial fusion protein Mfn2 causes mitochondrial dysfunction and cell death. *European Journal of Cell Biology* 86: 289–302
- Huo Y, Sun W, Shi T, Gao S & Zhuang M (2022) The MFN1 and MFN2 mitofusins promote clustering between mitochondria and peroxisomes. *Commun Biol* 5: 1–11
- Justine Pascual, Jelle Jacobs, Leticia Sansores-Garcia, Malini Natarajan, Julia Zeitlinger, Stein Aerts, Georg Halder FH (2017) Hippo reprograms the transcriptional response to Ras signaling. *Developmental Cell* 42: 1–28
- Kiel C, Filchtinski D, Spoerner M, Schreiber G, Kalbitzer HR & Herrmann C (2009) Improved binding of raf to Ras.GDP is correlated with biological activity. *Journal of Biological Chemistry* 284: 31893–902
- Killoran RC & Smith MJ (2019) Conformational resolution of nucleotide cycling and effector interactions for multiple small GTPases determined in parallel. *Journal of Biological Chemistry* 294: 9937–9948
- Lee J-Y, Nagano Y, Taylor JP, Lim KL & Yao T-P (2010) Disease-causing mutations in Parkin impair mitochondrial ubiquitination, aggregation, and HDAC6-dependent mitophagy. *Journal of Cell Biology* 189: 671–679
- Lin L & Bivona TG (2016) The Hippo effector YAP regulates the response of cancer cells to MAPK pathway inhibitors. *Molecular & cellular oncology* 3: e1021441
- Manning G, Young SL, Miller WT & Zhai Y (2008) The protist, *Monosiga brevicollis*, has a tyrosine kinase signaling network more elaborate and diverse than found in any known metazoan. *Proc Natl Acad Sci USA* 105: 9674–9679
- Ohba Y, Mochizuki N, Matsuo K, Yamashita S, Nakaya M, Hashimoto Y, Hamaguchi M, Kurata T, Nagashima K & Matsuda M (2000) Rap2 as a Slowly Responding Molecular Switch in the Rap1 Signaling Cascade. *Molecular and Cellular Biology* 20: 6074–6083
- Paganini S, Guidetti GF, Catricalà S, Trionfini P, Panelli S, Balduini C & Torti M (2006) Identification and biochemical characterization of Rap2C, a new member of the Rap family of small GTP-binding proteins. *Biochimie* 88: 285–295
- Patel A, Lee HO, Jawerth L, Maharana S, Jahnel M, Hein MY, Stoynev S, Mahamid J, Saha S, Franzmann TM, *et al* (2015) A Liquid-to-Solid Phase Transition of the ALS Protein FUS Accelerated by Disease Mutation. *Cell* 162: 1066–1077

- Peters D, Kay L, Eswaran J, Lakey J & Soundararajan M (2018) Human Miro Proteins Act as NTP Hydrolases through a Novel, Non-Canonical Catalytic Mechanism. *International Journal of Molecular Sciences* 19: 3839
- Quirion L, Robert A, Boulais J, Huang S, Astrain GB, Strakhova R, Jo CH, Kherdjemil Y, Faubert D, Thibault M-P, *et al* (2024) Mapping the global interactome of the ARF family reveals spatial organization in cellular signaling pathways. *Journal of Cell Science*
- Ray S, Singh N, Kumar R, Patel K, Pandey S, Datta D, Mahato J, Panigrahi R, Navalkar A, Mehra S, *et al* (2020) α -Synuclein aggregation nucleates through liquid–liquid phase separation. *Nat Chem* 12: 705–716
- Sebé-Pedrós A, Zheng Y, Ruiz-Trillo I & Pan D (2012) Premetazoan Origin of the Hippo Signaling Pathway. *Cell Reports* 1: 13–20
- Shai N, Yifrach E, van Roermund CWT, Cohen N, Bibi C, IJlst L, Cavellini L, Meurisse J, Schuster R, Zada L, *et al* (2018) Systematic mapping of contact sites reveals tethers and a function for the peroxisome-mitochondria contact. *Nat Commun* 9: 1761
- Shao DD, Xue W, Krall EB, Bhutkar A, Piccioni F, Wang X, Schinzel AC, Sood S, Rosenbluh J, Kim JW, *et al* (2014) KRAS and YAP1 converge to regulate EMT and tumor survival. *Cell* 158: 171–184
- Slemmons KK, Crose LES, Rudzinski E, Bentley RC & Linardic CM (2015) Role of the YAP Oncoprotein in Priming Ras-Driven Rhabdomyosarcoma. *Plos One* 10: e0140781
- Sloat SR & Hoppins S (2023) A dominant negative mitofusin causes mitochondrial perinuclear clusters because of aberrant tethering. *Life Sci Alliance* 6: e202101305
- Smith KP, Focia PJ, Chakravarthy S, Landahl EC, Klosowiak JL, Rice SE & Freymann DM (2020) Insight into human Miro1/2 domain organization based on the structure of its N-terminal GTPase. *J Struct Biol* 212: 107656
- Smith MJ & Ikura M (2014) Integrated RAS signaling defined by parallel NMR detection of effectors and regulators. *Nature Chemical Biology* 10: 223–230
- Smith MJ, Neel BG & Ikura M (2013) NMR-based functional profiling of RASopathies and oncogenic RAS mutations. *Proceedings of the National Academy of Sciences of the United States of America* 110: 4574–4579
- Smith MJ, Ottoni E, Ishiyama N, Goudreault M, Haman A, Meyer C, Tucholska M, Gasmi-Seabrook G, Menezes S, Laister RC, *et al* (2017) Evolution of AF6-RAS association and its implications in mixed-lineage leukemia. *Nature Communications* 8: 1–13
- Stiegler AL & Boggon TJ (2020) The pseudoGTPase group of pseudoenzymes. *The FEBS Journal* 287: 4232–4245

- Thillaivillalan D, Singh S, Killoran R, Singh A, Xu X, Shifman J & Smith MJ (2020) RASSF effectors couple diverse RAS subfamily GTPases to the Hippo pathway. *Science Signaling* 13: 1–15
- Tu B, Yao J, Ferri-Borgogno S, Zhao J, Chen S, Wang Q, Yan L, Zhou X, Zhu C, Bang S, *et al* (2019) YAP1 oncogene is a context-specific driver for pancreatic ductal adenocarcinoma. *JCI Insight* 4: 130811
- Uechi Y, Bayarjargal M, Umikawa M, Oshiro M, Takei K, Yamashiro Y, Asato T, Endo S, Misaki R, Taguchi T, *et al* (2009) Rap2 function requires palmitoylation and recycling endosome localization. *Biochemical and Biophysical Research Communications* 378: 732–737
- Wang L, Choi K, Su T, Li B, Wu X, Zhang R, Driskill JH, Li H, Lei H, Guo P, *et al* (2022) Multiphase coalescence mediates Hippo pathway activation. *Cell* 185: 4376-4393.e18
- Zhang W, Nandakumar N, Shi Y, Manzano M, Smith A, Graham G, Gupta S, Vietsch EE, Laughlin SZ, Wadhwa M, *et al* (2014) Downstream of mutant KRAS, the transcription regulator YAP is essential for neoplastic progression to pancreatic ductal adenocarcinoma. *Science Signaling* 7: ra42

Dear Dr. Smith

Thank you for the submission of your revised manuscript to EMBO reports. We have now received the full set of referee reports that is copied below.

As you will see, all referees are very positive about the study and request only minor changes to clarify text and figures.

From the editorial side, there are also a few things that we need before we can proceed with the official acceptance of your study.

- Please reduce the number of keywords to 5.
- Data availability paragraph: perhaps the text could be changed to "This study includes no data deposited in external repositories".
- Please update the 'Conflict of interest' paragraph to our new 'Disclosure and competing interests statement'. For more information see <https://www.embopress.org/page/journal/14693178/authorguide#conflictsofinterest>
- Regarding the Author Contributions, we now use CRediT to specify the contributions of each author in the journal submission system. Therefore, please remove the Author Contributions from the manuscript file and make sure that the author contributions in our manuscript tracking system are correct and up-to-date. The information you specified in the system will be automatically retrieved and typeset into the article. You can enter additional information in the free text box provided, if you wish.
- 'Materials and Methods' should be 'Methods'.
- Please describe your findings in the Abstract in present tense.
- We perform a routine image analysis on all manuscripts prior to acceptance. In the course of this analysis we noted the following aberrations, that I kindly ask you to check and clarify:

A) It seems that the Western blots shown in Figure 1C (left) are identical with the blots shown in Appendix Fig S3 (second row from top, right-most blot, Lysate and pull-down). The same applies to the Western blots in 1C (left) and the blots in S3 (fourth row from top, left-most, Lysate and pull-down).

B) The immunofluorescence images for RASD1-S42V shown in Figure 5D seem to have been reused in Fig. EV4C.

C) The lysate blots in Appendix Fig S1 (second row from top, left-most) and that in Appendix Fig S4 (second row from top, left-most) appear to be the same, even though the experimental conditions appear different in this case.

D) The pulldown blots in Appendix Fig S4 second row from top, right-most and third row from top, left-most, appear to be the same, even though the lysate blots differ.

Please carefully check the composition of these panels. In case the re-use is intentional, please clearly state so in the Figure legends to avoid any ambiguities.

In case the experiments show data from replicates, I recommend showing another replicate instead. Please also provide the source data for the Appendix images.

- Short note regarding Source Data folders' names for Fig. 1C: ARF1.. should be labeled 'right' and HRAS... should be labeled 'left'
 - Our production/data editors have asked you to clarify several points in the figure legends (see below). Please incorporate these changes in the manuscript and return the revised file with tracked changes with your final manuscript submission.
- A) Figure legend text:
- Please note that the figure 5d is mislabeled as figure 5e in the manuscript. This needs to be rectified.
- B) Statistical test information. Only p-values that are actually shown in the figure panel(s) should (and must) be defined in the legends, all others should be removed from (or added to) the legend. Moreover, we ask for the specification of exact p-values:
- Please note that the exact p values are not provided in the legend of figure 4e.
 - Please indicate the statistical test used for data analysis in the legends of figures EV 5a.

C) Replicates and error bars:

- Although 'n' is provided, please describe the nature of entity for 'n' in the legends of figures EV 3c; EV 4d.
- Please note that for heatmap present in figure EV 1a; a numbered scale bar is not provided. This needs to be rectified.

- Finally, EMBO Reports papers are accompanied online by

A) a short (1-2 sentences) summary of the findings and their significance,

B) 2-3 bullet points highlighting key results and

C) a schematic summary figure that provides a sketch of the major findings (not a data image).

Please provide the summary figure as a separate file in PNG or JPG format at a size of 550x300-600 pixels (width x height).

Please note that the size is rather small and that text needs to be readable at the final size. Please send us this information along with the revised manuscript.

- On a different note, I would like to alert you that EMBO Press offers a new format for a video-synopsis of work published with us, which essentially is a short, author-generated film explaining the core findings in hand drawings, and, as we believe, can be very useful to increase visibility of the work. This has proven to offer a nice opportunity for exposure i.p. for the first author(s) of the study. Please see the following link for representative examples and their integration into the article web page:

<https://www.embopress.org/doi/full/10.15252/embo.2019103932>

With kind regards,

Martina

Referee #1:

The authors have dealt satisfactorily with most of the issues I raised and should be congratulated on their really hard work and great effort. The only aspect where I still have some degree of semantic disagreement with the authors is the definition of canonical GTPase effectors and the GDP/GTP dependence of interactions. The authors point out that the canon/prototype as established based on the K/H/N-Ras findings of the 1980s does not necessarily apply to all the other 150 GTPases as we know them today, and they give some examples, including MIRO. I fully agree with the authors that there are certainly exceptions to the canon, such as possibly MIRO, and we certainly need to revisit it. Therefore, I do not think it is appropriate to describe RASSF3 as a canonical effector of MIRO, as the authors do in the abstract. This interaction is nucleotide independent, as the authors show in Fig2D. According to the canon, this would rather apply to a GTPase regulator, GDI or associated protein upstream of MIRO. So while I agree that the canon should be revisited at some point, I would strongly recommend changing the wording in the abstract. I would recommend changing the wording in the abstract: "The poorly studied RASSF3 was identified as the first canonical effector of mitochondrial MIRO,..." for "potential effector", just "effector" or even "non-canonical effector" as the current canon implies GTP dependence in the interaction, which is not the case here. Apart from this, I support this manuscript for publication.

Referee #2:

Authors have addressed properly my concerns/observations.

Additional small observations:

- Introduction, Page 3: "... leaving the remaining 9 RASSF effectors with no known GTPase binding partner". As article data indicates too, perhaps effectors have to be changed "9 family members that function either as effectors or interacting partners for GTPases" or similar.
- Abbreviations have to be checked one more throughout and introduced where they appear first time. E.g., YAP, Yes-Associated Protein (Yap); LATS1/2, large tumor suppressor, homolog 1/2, etc.
- Results: e.g., Fig 1A has to be Fig. 1A, using a dot after Fig. Similar for the rest of the text where figures are called.

Reviewer Notes

Referee #1:

The authors have dealt satisfactorily with most of the issues I raised and should be congratulated on their really hard work and great effort. The only aspect where I still have some degree of semantic disagreement with the authors is the definition of canonical GTPase effectors and the GDP/GTP dependence of interactions. The authors point out that the canon/prototype as established based on the K/H/N-Ras findings of the 1980s does not necessarily apply to all the other 150 GTPases as we know them today, and they give some examples, including MIRO. I fully agree with the authors that there are certainly exceptions to the canon, such as possibly MIRO, and we certainly need to revisit it. Therefore, I do not think it is appropriate to describe RASSF3 as a canonical effector of MIRO, as the authors do in the abstract. This interaction is nucleotide independent, as the authors show in Fig2D. According to the canon, this would rather apply to a GTPase regulator, GDI or associated protein upstream of MIRO. So while I agree that the canon should be revisited at some point, I would strongly recommend changing the wording in the abstract. I would recommend changing the wording in the abstract: "The poorly studied RASSF3 was identified as the first canonical effector of mitochondrial MIRO,..." for "potential effector", just "effector" or even "non-canonical effector" as the current canon implies GTP dependence in the interaction, which is not the case here. Apart from this, I support this manuscript for publication.

We thank the referee for these comments, and at their suggestion have changed the wording in the abstract from “canonical effector” to “potential effector”.

Referee #2:

Authors have addressed properly my concerns/observations.

Additional small observations:

- Introduction, Page 3: "... leaving the remaining 9 RASSF effectors with no known GTPase binding partner". As article data indicates too, perhaps effectors have to be changed "9 family members that function either as effectors or interacting partners for GTPases" or similar.*
- Abbreviations have to be checked one more throughout and introduced where they appear first time. E.g., YAP, Yes-Associated Protein (Yap); LATS1/2, large tumor suppressor, homolog 1/2, etc.*
- Results: e.g., Fig 1A has to be Fig. 1A, using a dot after Fig. Similar for the rest of the text where figures are called.*

We have changed the reference to “RASSF effectors” in the introduction to “RASSF proteins”, and all callouts to “Fig.” For the abbreviations, we have left these but can add additional information if required by the editorial team.

Editorial Notes

- Please reduce the number of keywords to 5.*

Done.

- *Data availability paragraph: perhaps the text could be changed to "This study includes no data deposited in external repositories".*

This was changed.

- *Please update the 'Conflict of interest' paragraph to our new 'Disclosure and competing interests statement'. For more information see*

<https://www.embopress.org/page/journal/14693178/authorguide#conflictofinterest>

This has been updated.

- *Regarding the Author Contributions, we now use CRediT to specify the contributions of each author in the journal submission system. Therefore, please remove the Author Contributions from the manuscript file and make sure that the author contributions in our manuscript tracking system are correct and up-to-date. The information you specified in the system will be automatically retrieved and typeset into the article. You can enter additional information in the free text box provided, if you wish.*

This has been completed and the section removed from the manuscript.

- *'Materials and Methods' should be 'Methods'.*

This is corrected.

- *Please describe your findings in the Abstract in present tense.*

This has been updated.

- *We perform a routine image analysis on all manuscripts prior to acceptance. In the course of this analysis we noted the following aberrations, that I kindly ask you to check and clarify:*

A) I seems that the Western blots shown in Figure 1C (left) are identical with the blots shown in Appendix Fig S3 (second row from top, right-most blot, Lysate and pull-down). The same applies to the Western blots in 1C (left) and the blots in S3 (fourth row from top, left-most, Lysate and pull-down).

This was initially done intentionally, but we have now changed the blots in the Appendix section to a second replicate of the same experiment.

B) The immunofluorescence images for RASD1-S42V shown in Figure 5D seem to have been reused in Fig. EV4C.

This was done intentionally to demonstrate no bleed through when using these channels. We have added clarification to the Figure legend (5D).

C) The lysate blots in Appendix Fig S1 (second row from top, left-most) and that in Appendix Fig S4 (second row from top, left-most) appear to be the same, even though the experimental conditions appear different in this case.

D) The pull-down blots in Appendix Fig S4 second row from top, right-most and third row from top, left-most, appear to be the same, even though the lysate blots differ.

These were unintentionally mixed during composition of the Figures, and we have replaced with the proper panels.

Please carefully check the composition of these panels. In case the re-use is intentional, please clearly state so in the Figure legends to avoid any ambiguities.

In case the experiments show data from replicates, I recommend showing another replicate instead. Please also provide the source data for the Appendix images.

- Short note regarding Source Data folders' names for Fig. 1C: ARF1.. should be labeled 'right' and HRAS... should be labeled 'left'

This has been corrected.

- Our production/data editors have asked you to clarify several points in the figure legends (see below). Please incorporate these changes in the manuscript and return the revised file with tracked changes with your final manuscript submission.

A) Figure legend text:

- Please note that the figure 5d is mislabeled as figure 5e in the manuscript. This needs to be rectified.

Corrected.

B) Statistical test information. Only p-values that are actually shown in the figure panel(s) should (and must) be defined in the legends, all others should be removed from (or added to) the legend. Moreover, we ask for the specification of exact p-values:

- Please note that the exact p values are not provided in the legend of figure 4e.

The exact p values have been added directly to the figure, and are now specified in the Figure legend as being <0.0001 .

- Please indicate the statistical test used for data analysis in the legends of figures EV 5a.

This statistical analysis was performed by Metascape, and relies on a “hypergeometric test and Benjamini-Hochberg p-value correction algorithm” (Zhou *et al.* 2019 *Nat Comm*). We have added this information to the Figure Legends of Figs EV3 and EV5.

C) Replicates and error bars:

- Although 'n' is provided, please describe the nature of entity for 'n' in the legends of figures EV 3c; EV 4d.

These are biological replicates, now explained in the legends.

- Please note that for heatmap present in figure EV 1a; a numbered scale bar is not provided. This needs to be rectified.

Numbers have been added.

*- Finally, EMBO Reports papers are accompanied online by
A) a short (1-2 sentences) summary of the findings and their significance,*

Effectors in the RASSF family demonstrate distinct and wide-ranging specificity for RAS superfamily GTPases. Identified complexes can regulate Hippo signalling and control nuclear shuttling of YAP, as well as additional cellular processes such as mitochondrial distribution.

B) 2-3 bullet points highlighting key results and

- The RA domains of RASSF proteins complex with diverse RAS superfamily GTPases
- RASSF-GTPase interactions alter YAP subcellular distribution, including its sequestration in membraneless organelles
- RASSF3 is recruited to mitochondria by MIRO GTPases and they impact distribution of mitochondria and peroxisomes

*C) a schematic summary figure that provides a sketch of the major findings (not a data image).
Please provide the summary figure as a separate file in PNG or JPG format at a size of 550x300-600 pixels (width x height). Please note that the size is rather small and that text needs to be readable at the final size. Please send us this information along with the revised manuscript.*

This has been uploaded.

Dr. Matthew Smith
Université de Montréal
Institute for Research in Immunology and Cancer (IRIC)
2950 chemin de Polytechnique
Montreal, Quebec H3T 1J4
Canada

Dear Dr. Smith,

I am very pleased to accept your manuscript for publication in the next available issue of EMBO reports. Thank you for your contribution to our journal.

Yours sincerely,
